# On the benefits of the tryptophan metabolite 3-hydroxyanthranilic acid in *Caenorhabditis elegans* and mouse aging

Hope Dang[1,3], Raul Castro-Portuguez[1,3], Luis Espejo[1,3], Grant Backer[2], Samuel Freitas [1], Erica Spence[1], Jeremy Meyers[1], Karissa Shuck [1], Emily A. Gardea[1], Leah M. Chang[1], Jonah Balsa[1], Niall Thorns[1], Caroline Corban [2], Teresa Liu [2], Shannon Bean[2], Susan Sheehan [2], Ron Korstanje [2,4] & George L. Sutphin [1,4] ✉

Tryptophan metabolism through the kynurenine pathway influences molecular processes critical to healthy aging including immune signaling, redox homeostasis, and energy production. Aberrant kynurenine metabolism occurs during normal aging and is implicated in many age-associated pathologies including chronic inflammation, atherosclerosis, neurodegeneration, and cancer. We and others previously identified three kynurenine pathway genes—*tdo-2*, *kynu-1*, and *acsd-1*—for which decreasing expression extends lifespan in invertebrates. Here we report that knockdown of *haao-1*, a fourth gene encoding the enzyme 3-hydroxyanthranilic acid (3HAA) dioxygenase (HAAO), extends lifespan by ~30% and delays age-associated health decline in *Caenorhabditis elegans*. Lifespan extension is mediated by increased physiological levels of the HAAO substrate 3HAA. 3HAA increases oxidative stress resistance and activates the Nrf2/SKN-1 oxidative stress response. In pilot studies, female *Haao* knockout mice or aging wild type male mice fed 3HAA supplemented diet were also long-lived. HAAO and 3HAA represent potential therapeutic targets for aging and age-associated disease.

Physiological tryptophan (TRP) has a variety of potential fates in eukaryotic cells, including incorporation into proteins, conversion to serotonin or melatonin, conversion to tryptamine, or conversion to nicotinamide adenine dinucleotide (NAD$^+$) through the kynurenine pathway (Fig. 1a). The majority (>90%) of ingested tryptophan is catabolized by the kynurenine pathway in mammals[1]. Altered kynurenine pathway activity has been implicated in a range of age-associated diseases in humans, including cardiovascular disease, kidney disease, cancer, and neurodegeneration[2,3]. The structure of the kynurenine pathway is evolutionarily conserved from bacteria through mammals, with a few notable exceptions. The initial, rate-limiting conversion of TRP to N-formylkynurenine (NFK) is carried out by the enzyme tryptophan 2,3-dioxygenase (TDO2) across species. Vertebrate genomes encode a second cytokine-responsive enzyme, IDO1, that catalyzes this reaction, and vertebrate kynurenine pathway initiation is largely segregated between immune cells (IDO1) and liver (TDO2)[4]. A duplication event in the mammalian lineage produced IDO2, which is more ubiquitously expressed and less active than IDO1 and does not respond to cytokines. IDO2's function is currently less well-defined[4]. Arylformamidase (AFMID) catabolizes NFK to kynurenine (KYN), which is in turn processed to either kynurenic acid (KA) or NAD$^+$ through the two major branches of the kynurenine pathway. The NAD$^+$ branch converts KYN to NAD$^+$ through a series of metabolic steps catalyzed by the enzymes

[1]Molecular & Cellular Biology, University of Arizona, Tucson, AZ, USA. [2]The Jackson Laboratory, Bar Harbor, ME, USA. [3]These authors contributed equally: Hope Dang, Raul Castro-Portuguez, Luis Espejo. [6]These authors jointly supervised this work: Ron Korstanje, George L. Sutphin. ✉e-mail: sutphin@arizona.edu

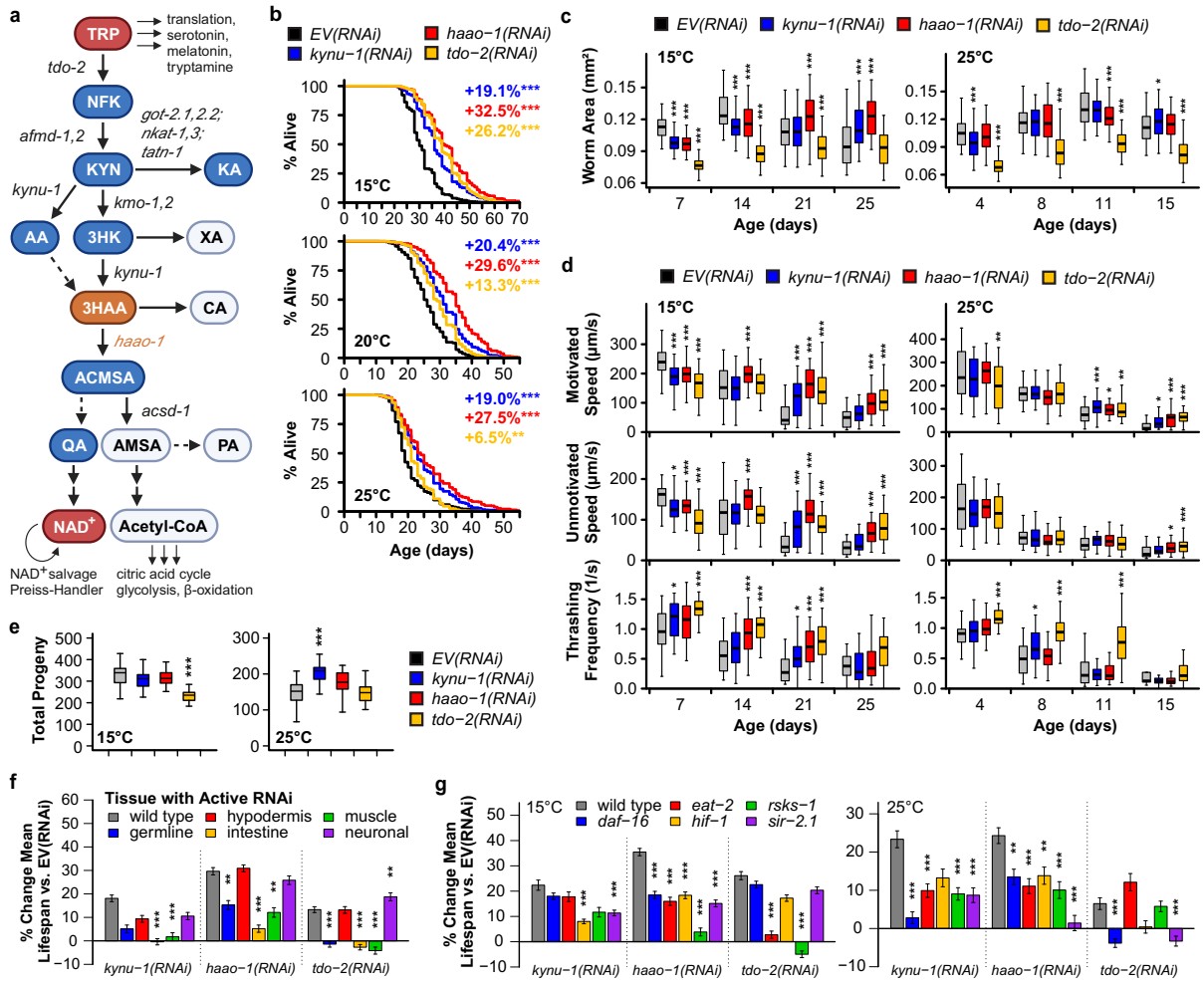

**Fig. 1 | Knockdown of *haao-1* extends healthy lifespan in *C. elegans*.**
**a** Kynurenine pathway schematic (created with BioRender). Metabolites are shown in bubbles. *C. elegans* gene names for enzymes are shown in italics. RNAi knockdown of *kynu-1*, *haao-1*, or *tdo-2*: **b** extends lifespan at 15 °C, 20 °C, and 25 °C (colored text indicates change in mean lifespan relative to empty vector (EV) RNAi); **c** slows growth but allows worms to maintain body size (i.e., worm area) later in life; and **d** slows motivated and unmotivated crawl speed early in life but allows worms to maintain speed and thrashing frequency in liquid later in life. For box-and-whisker plots, center line represents median, boxes indicate first and third quartiles, and whiskers represent 5th and 95th percentiles. **e** *tdo-2(RNAi)* reduces brood size at 15 °C while *kynu-1(RNAi)* increased brood size at 25 °C. Lifespan extension

from *kynu-1(RNAi)*, *haao-1(RNAi)*, or *tdo-2(RNAi)* displays different patterns of dependence on (**f**) tissue and (**g**) genes in established aging pathways. Error bars indicate standard error of mean. *$p < 0.05$, **$p < 0.01$, ***$p < 0.001$ (log rank test, panels **b**, **f**, **g**; two-sided Welch's *t* test, panels **c–e**). Abbreviations: TRP tryptophan, NFK N-formylkynurenine, KYN kynurenine, KA kynurenic acid, AA anthranilic acid, 3HK 3-hydroxykynurenine, XA xanthurenic acid, 3HAA 3-hydroxyanthranilic acid, CA cinnabarinic acid, ACMSA aminocarboxymuconate semialdehyde, AMSA aminomuconic semialdehyde, QA quinolinic acid, Acetyl-CoA acetyl coenzyme A, NAD+ nicotinamide adenine dinucleotide. Summary statistics are provided in Supplementary Data 1 and source data are provided in Source Data.

kynurenine 3-monooxygenase (KMO), kynureninase (KYNU), and 3-hydroxyanthranilic acid dioxygenase (HAAO).

Multiple kynurenine pathway interventions are beneficial to aging invertebrates. Oxenkrug[5] observed that *Drosophila melanogaster* strains with altered eye color resulting from mutations in the gene *vermillion*, encoding the initial kynurenine pathway enzyme TDO2, were long-lived, and later reported lifespan extension in wild type *Drosophila* treated with the TDO2 inhibitors α-methyltryptophan (αMT) or 5-methyltryptophan (5MT)[6]. In *Caenorhabditis elegans*, van der Goot et al.[7] found that knockdown of the *vermillion* ortholog, *tdo-2*, extended lifespan and improved α-synuclein pathology, likely mediated by elevated TRP[7]. We subsequently discovered that knockdown of *kynu-1*, encoding KYNU, extended healthy *C. elegans* lifespan[8] through mechanisms distinct from *tdo-2* knockdown[7]. Finally, Katsyuba et al.[9] recently reported lifespan extension following knockdown of *acsd-1*, encoding aminocarboxymuconate semialdehyde decarboxylase

(ACMSD), which controls one branchpoint of the kynurenine pathway that can lead alternately to production of picolinic acid and acetyl coenzyme-A, or to quinolinic acid (QA), a precursor for nicotinamide adenine dinucleotide (NAD+) (Fig. 1a). Reduced ACMSD activity elevated NAD+ production, the proposed mediator of lifespan extension[9]. Knockdown of *acsd-1* has the opposite effect on physiological NAD+ levels compared to knockdown of either *tdo-2* or *kynu-1*, implying that kynurenine metabolism can influence aging through both NAD+-dependent and NAD+-independent mechanisms.

Given that three genes in the kynurenine pathway had been implicated in aging through distinct mechanisms[7–9], we asked whether other targets in the kynurenine pathway may also impart benefits in *C. elegans*. In vertebrate and most invertebrate model systems, three kynurenine pathway enzymes aside from TDO2, KYNU, and ACMSD are encoded by single genes and thus amenable to straightforward knockdown using RNAi: AFMID, KMO, and HAAO. However,

duplication events in the nematode lineage resulted in two genes encoding both AFMID and KMO (Fig. 1a)[10,11]. We therefore focused on HAAO.

Here we report that knockdown of the gene encoding HAAO, *haao-1*, robustly extended *C. elegans* lifespan by increasing physiological levels of the HAAO substrate metabolite 3-hydoxyanthranilic acid (3HAA), a mechanism distinct from inhibition of *tdo-2*, *kynu-1*, or *acsd-1*[7–9]. We further present evidence implicating heightened Nrf2/SKN-1-mediated oxidative stress resistance in the beneficial effects of 3HAA. Finally, we report that dietary supplementation with 3HAA starting late in life or lifelong knockout of *Haao* both extend lifespan in two small mouse studies.

## Results

### HAAO promotes *C. elegans* aging

We initially asked whether knockdown of *haao-1* would result in lifespan extension similar to *tdo-2*, *kynu-1*, or *acsd-1*[7–9]. Indeed, RNAi knockdown of *haao-1* robustly extended lifespan by ~30%, exceeding lifespan extension from either *kynu-1(RNAi)* or *tdo-2(RNAi)* (Fig. 1b). Like *kynu-1(RNAi)* but unlike *tdo-2(RNAi)*[8], this benefit was largely consistent across the viable temperature range for *C. elegans* (i.e., while *haao-1(RNAi)* worms experience a slightly higher lifespan extension at lower temperatures, the impact of temperature on the degree of lifespan extension is much larger for *tdo-2(RNAi)*). At 15 °C, *haao-1(RNAi)* animals had a similar peak body size to animals subjected to empty vector (EV) RNAi but were slow to reach full body size (day 21 vs. day 14) and resistant to the age-dependent reduction in body size observed in *EV(RNAi)* animals (Fig. 1c). At 25 °C, *haao-1(RNAi)* modestly reduced peak body size relative to *EV(RNAi)* but did not affect early or late body size. At 15 °C, *haao-1(RNAi)* animals displayed reduced motivated and unmotivated crawling speed on solid media relative to *EV(RNAi)* animals during early adulthood but were resistant to age-dependent reduction in motivated speed, unmotivated speed, and thrashing in liquid media (Fig. 1d). These health benefits were blunted at 25 °C, resembling the pattern for *kynu-1(RNAi)* but distinct from *tdo-2(RNAi)* (Fig. 1c, d). Total brood size was not impacted by *haao-1(RNAi)* (Fig. 1e); however, egg production was slightly delayed relative to *EV(RNAi)*, similar to *kynu-1(RNAi)* and less pronounced than *tdo-2(RNAi)* (Supplementary Fig. 1a).

We next measured the impact of *kynu-1(RNAi)*, *haao-1(RNAi)*, and *tdo-2(RNAi)* on lifespan in a set of *C. elegans* strains with molecular machinery for RNAi globally inactivated and reactivated in restricted tissues, allowing tissue-specific RNAi knockdown. For all three genes, RNAi knockdown in either the hypodermis or neurons extended lifespan to a similar degree as whole-body knockdown. RNAi knockdown in intestine or muscle was insufficient to extend lifespan for *kynu-1* or *tdo-2* and produced intermediate lifespan extension for *haao-1* (Fig. 1f, Supplementary Fig. 2). Germline-specific knockdown generated a similar pattern to intestine and muscle, except that variability between replicates made the outcome indeterminate for *kynu-1(RNAi)*. This pattern suggests that the benefits of kynurenine pathway interventions in *C. elegans* are likely driven by activity in hypodermis, neurons, and possibly other tissues not examined.

We next examined the interaction between kynurenine metabolism and genetic pathways with established roles in aging by measuring the impact of *kynu-1(RNAi)*, *haao-1(RNAi)*, and *tdo-2(RNAi)* on lifespan in strains with the following mutations: *daf-16(mu68)* (impaired insulin/IGF-1-like signaling), *eat-2(ad465)* (reduced pharyngeal pumping, modeling dietary restriction), *hif-1(ia4)* (impaired hypoxic response), *rsks-1(ok1255)* (impaired translation regulation in response to mTOR signaling), and *sir-2.1(ok434)* (*C. elegans* Sir2 ortholog knockout). As previously reported, *kynu-1* and *tdo-2* displayed distinct and temperature-dependent patterns of interaction with this set of mutations[8]. The pattern of interaction for *haao-1* was distinct from either gene. Lifespan extension from *haao-1(RNAi)* was reduced in all

genetic backgrounds at both 15 °C and 25 °C relative to wild type worms and absent in *rsks-1(ok1255)* worms at 15 °C and *sir-2.1(ok434)* worms at 25 °C (Fig. 1g, Supplementary Figs. 3, 4). This pattern suggests a complex relationship with established aging pathways and does not clearly implicate a single primary mediator. Given this data, we shifted our focus away from working "up" from established aging pathways to instead work "down" from the functional role of HAAO.

### 3HAA mediates *haao-1* lifespan extension

In our lifespan experiments we noted that animals with reduced *haao-1* activity developed a red coloration that became visible near the head under brightfield illumination around day 4 of adulthood and throughout the body with age (Fig. 2a, Supplementary Fig. 5a). The structure of the kynurenine pathway suggests that inhibiting HAAO should result in physiological accumulation of its substrate metabolite, 3HAA (Fig. 1a). 3HAA proved to be red in color (Fig. 2b), consistent with 3HAA accumulation causing the red coloration in worms lacking active *haao-1*. Liquid chromatography tandem massspectrometry (LC-MS/MS) confirmed that 3HAA is elevated ~10-fold in worms subjected to *haao-1(RNAi)*, but not *kynu-1(RNAi)* or *tdo-2*(RNAi) (Fig. 2c), and that this accumulation increases with age (Supplementary Fig. 6a). We hypothesized that 3HAA accumulation mediates lifespan extension from *haao-1* knockdown. In support of this hypothesis, we found that worms maintained on solid nematode growth media (NGM) supplemented with 0.01 to 2.5 mM 3HAA were long-lived relative to untreated animals (Fig. 2d, Supplementary Fig. 6l, m). 1 mM 3HAA closely mimicked lifespan extension from *haao-1(RNAi)* or the null mutation *haao-1(tm4627)* (Fig. 2e). Treating worms with 0.01 to 10 µM 4CL-3HAA, a 3HAA analog and HAAO inhibitor[12,13], also increased *C. elegans* lifespan in a dose-dependent manner (Supplementary Fig. 6n). While lifespan extension was less for 4CL-3HAA than *haao-1(RNAi)*, we observed the largest effect at the lowest dose tested (0.01 µM), and toxicity at the highest (1 mM), suggesting that we have not identified the optimal dose for longevity. 1 mM 3HAA, like deletion or knockdown of *haao-1*, did not affect brood size (Fig. 2f, Supplementary Fig. 5b) and slowed age-associated decline in thrashing in liquid (Fig. 2g). Both *haao-1(RNAi)* and 1 mM 3HAA extended lifespan to a similar degree when initiated from egg or from the first day of adulthood, indicating that the impact of these interventions during development is not necessary for lifespan extension (Supplementary Fig. 5c–e). If our model is accurate and 3HAA mediates benefits of decreasing activity of HAAO, but not KYNU or TDO2, then 3HAA should further increase the already long lifespan of animals with reduced *kynu-1* or *tdo-2*, but not animals with reduced *haao-1*. Our observations confirmed this prediction (Fig. 2h, Supplementary Fig. 7a). 1 mM 3HAA did slightly extended lifespan in *haao-1(tm4627)* animals, which we speculate results from increased earlylife 3HAA exposure. Supporting this notion and further validating our model that 3HAA mediates the health benefits from *haao-1* knockdown, full lifespan extension from 3HAA supplementation was rescued in *haao-1(tm4627)* animals when 3HAA production was blocked by deletion of *kynu-1* (Figs. 1a, 2a, Supplementary Fig. 7a). Another non-mutually alternative model is that reduced HAAO activity and/or elevated 3HAA produces feedback on the kynurenine pathway by inhibiting KYNU or TDO2, and that lifespan extension is mediated by up- or downregulation of a metabolite upstream of 3HAA. We observed an increase in both TRP and KA in response to knockdown of all three genes at 15 °C, but this effect did not replicate in *kynu-1* and *haao-1* knockout animals at 20 °C (Supplementary Fig. 6b, c). Knockdown or knockout of *haao-1* did produced a modest (~2-fold) and consistent increase in KYN; however, the levels were lower than that produced by *kynu-1* knockdown (~10-fold; Supplementary Fig. 6b, c), were not maintained with age (Supplementary Fig. 6c), and supplementing animals with 0.01 to 1 mM KYN slightly shortened lifespan (Supplementary Fig. 6o). Simultaneous knockdown of *haao-1*

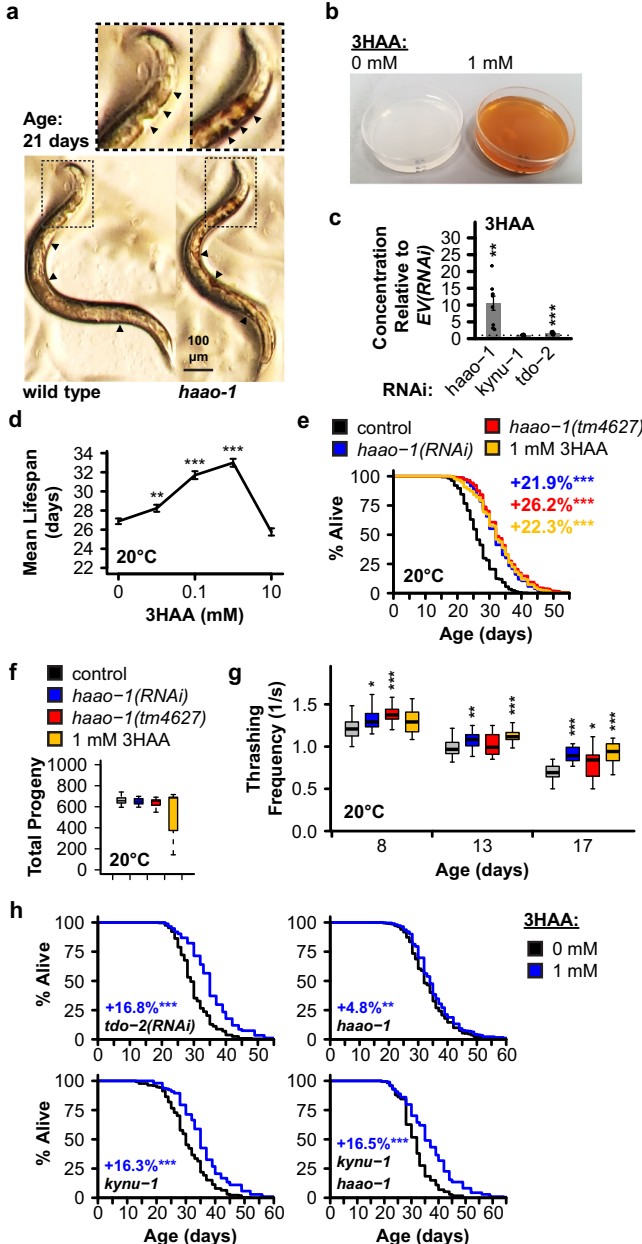

**Fig. 2 | 3HAA mediates lifespan extension from reduced *haao-1* activity. a** *C. elegans* lacking *haao-1* (*haao-1(tm4637)*) accumulate a red coloration with age relative to wild type. Animals shown are age 21 days from egg. Cutout provides an enlarged view of the head region. Arrows highlight areas of distinct coloration difference between wild type and *haao-1* animals. **b** NGM plates supplemented with 3HAA are red in color. **c** 3HAA levels measured by LC-MS/MS are elevated ~10-fold in 8-day-old worms subjected to *haao-1(RNAi)*, but not *kynu-1(RNAi)* or *tdo-2(RNAi)*, relative to *EV(RNAi)* (N = 9 biologically independent samples per condition). **d** Supplementing *C. elegans* media with 0.01 to 1 mM 3HAA extends lifespan. **e** 1 mM 3HAA mimics lifespan extension from *haao-1* knockdown or deletion and similarly **f** does not affect brood size and **g** elevates thrashing capacity in liquid across lifespan. For box-and-whisker plots, center line represents median, boxes indicate first and third quartiles, and whiskers represent 5th and 95th percentiles. **h** 1 mM 3HAA increases lifespan in animals subjected to *tdo-2(RNAi)*, *kynu-1(tm4924)* single deletion, or *kynu-1(tm4924) haao-1(tm4637)* double deletion to a degree similar to wild type animals; lifespan extension is largely eliminated in *haao-1(tm4637)* animals. Colored text indicates change in mean lifespan relative to control. Error bars indicate standard error of mean. *p < 0.05, **p < 0.01, ***p < 0.001 (two-sided Welch's *t* test, panels **c**, **f**, **g**; log rank test, panels **d**, **e**, **h**). Summary statistics are provided in Supplementary Data 1 and source data are provided in Source Data.

and *kynu-1* also produces lifespan extension more similar to *kynu-1* (Supplementary Fig. 7b–d). In contrast, knockdown of *tdo-2*, *kynu-1*, and *haao-1* all produced a ~4- to 10-fold reduction in AA at 15 °C that replicated in *haao-1* and *kynu-1* knockout animals and was maintained with age at 20 °C (Supplementary Fig. 6e, k). Furthermore, supplementing animals with 0.01 to 10 mM AA reduced lifespan in a dose-dependent manner (Supplementary Fig. 6p), suggesting that in addition to 3HAA as a private mechanism for *haao-1* knockdown, repression of AA may be a common mechanism of action mediating lifespan extension from knockdown of *tdo-2*, *kynu-1*, and *haao-1*.

## 3HAA promotes oxidative stress resistance

We next asked what mechanisms may mediate lifespan extension from 3HAA. Accumulation of reactive oxygen species (ROS) and dysregulation of cellular oxidative stress response are causally implicated in a variety of age-associated pathologies[14]. 3HAA is redox active, with pro- or antioxidant properties depending on physiological context, including pH and the presence of metal ions[15]. Both *haao-1* knockdown and 3HAA supplementation increased survival of worms exposed to the superoxide generator paraquat (Fig. 3a). We attempted to examine the impact of 3HAA on resistance to exogenous hydrogen peroxide ($H_2O_2$); however, $H_2O_2$ resulted in a distinct lightening of the red coloration in 3HAA supplemented media (Fig. 3b), suggesting a direct interaction between 3HAA and $H_2O_2$. We hypothesized that 3HAA directly degrades hydrogen peroxide. We incubated 3HAA and $H_2O_2$ at a range of concentrations and measured the resulting $H_2O_2$ concentration. 3HAA reduced measurable $H_2O_2$ in a concentration-dependent manner (Fig. 3c) independent of pH (Supplementary Fig. 8a). *C. elegans* release endogenously produced $H_2O_2$ into their environment[16,17]. We found that $H_2O_2$ release increased with age in wild type animals, and that this increase was partially repressed in animals lacking *haao-1* or supplemented with 3HAA (Fig. 3d). In contrast to this interaction with $H_2O_2$, we found that endogenous levels of $H_2O_2$ (measured by the $H_2O_2$-sensitive HyPer fluorophore[18]) were elevated in *haao-1(tm4627)* animals (Fig. 3e). This adds to the existing body of evidence that 3HAA can both promote or inhibit ROS production, likely depending on multiple factors including the cell type, cellular redox environment, and specific ROS species of interest[15].

We next asked whether, on balance, the change in ROS produced by 3HAA would activate the cellular oxidative stress response. We examined the impact of 3HAA on four transgenic fluorescent oxidative stress response reporters: green fluorescent protein (GFP) fused to SKN-1, the *C. elegans* ortholog of the Nrf2 oxidative stress response transcription factor (*skn-1::GFP*), two SKN-1 transcriptional reporters with GFP expression driven by promoters from SKN-1 target genes involved in glutathione production (*gcs-1p::GFP*, *gst-4p::GFP*), and a DAF-16 transcriptional reporter with GFP expression driven by the superoxide dismutase 3 promoter (*sod-3p::GFP*). *haao-1(RNAi)* and 3HAA robustly increased SKN-1::GFP expression and both SKN-1 transcriptional reporters (Fig. 3f, g). Recently published work has reported elevated NRF2 protein levels in cultured HT1080[19] and HeLa[20] cells by 50 to 200 μM 3HAA and we confirmed this observation for human SK-Hep1 and PANC-1 cells in response to 100 μM 3HAA (Supplementary Fig. 8c). *haao-1(RNAi)*, but not 3HAA, slightly activated expression of *sod-3p::GFP* (Supplementary Fig. 8b), consistent with our observation that *haao-1(RNAi)* can extend lifespan in the absence of *daf-16* (Fig. 1g). Finally, we asked whether activation of Nrf2/SKN-1 is required for lifespan extension from elevated 3HAA. Lifespan extension from either *haao-1(RNAi)* (Fig. 3h) or 1 mM 3HAA (Fig. 3i) was largely, but not completely, suppressed in *C. elegans* with the hypomorphic *skn-1(zj15)* mutation. We concluded that 3HAA has both direct and indirect antioxidant activity in vivo in the context of aging *C. elegans*, and that 3HAA activation of SKN-1 partially mediates the aging benefits of *haao-1* knockdown or 3HAA supplementation.

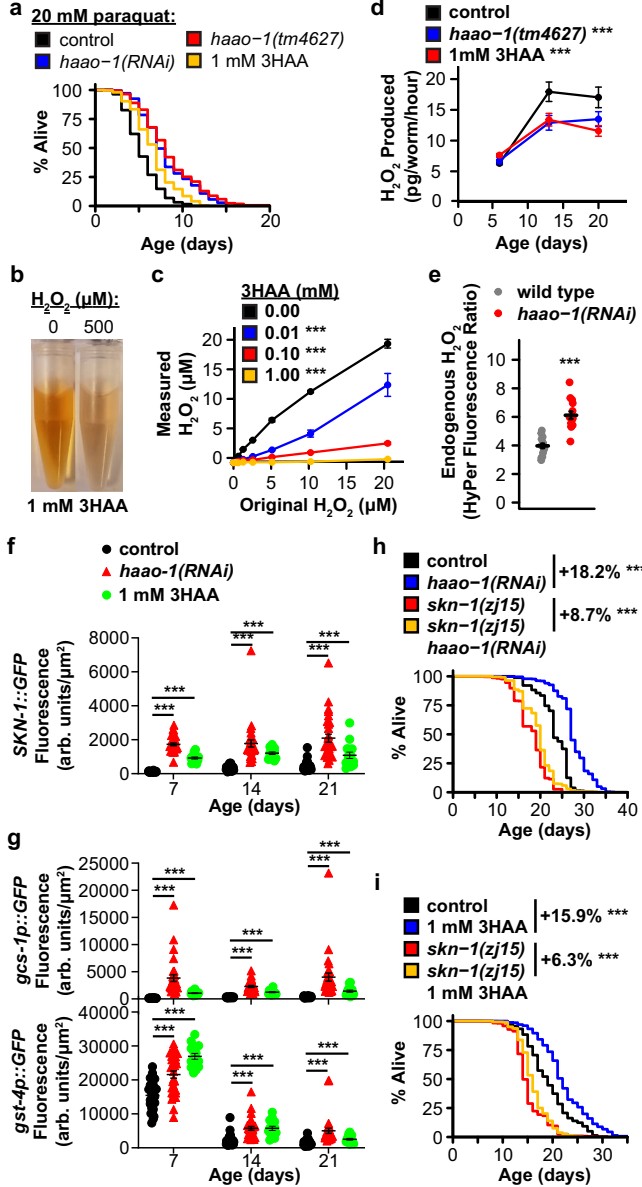

**Fig. 3 | Elevating 3HAA protects *C. elegans* against oxidative stress and activates the Nrf2/SKN-1 oxidative stress response. a** Knockdown or deletion of *haao-1* or 1 mM 3HAA increases survival of *C. elegans* challenged with 20 mM paraquat. **b** 500 μM $H_2O_2$ reduces visible red color of 1 mM 3HAA in water. **c** 3HAA reduces $H_2O_2$ in water in a dose-dependent manner. **d** Deletion of *haao-1* or 1 mM 3HAA represses the age-dependent increase in $H_2O_2$ secretion in *C. elegans*. **e** *C. elegans* lacking *haao-1* have elevated endogenous $H_2O_2$ (HyPer). *haao-1(RNAi)* or 1 mM 3HAA increases fluorescence in *C. elegans* strains transgenically expressing **f** the *skn-1::GFP* fusion protein or **g** the *gcs-1p::GFP* or *gst-4p::GFP* promoter activity reporters. Hypomorphic *skn-1* loss-of-function largely prevents lifespan extension from **h** *haao-1(RNAi)* or (**i**) 1 mM 3HAA. Colored text indicates change in mean lifespan relative to control. Error bars indicate standard error of mean. \*p < 0.05, \*\*p < 0.01, \*\*\*p < 0.001 (log rank test, panels **a**, **h**, **i**; ANOVA; panels **c**, **d**; Welch's *t* test, panels **f**, **g**). Summary statistics are provided in Supplementary Data 1 and source data are provided in Source Data.

## Dietary 3HAA and *Haao* knockout extend mouse lifespan

In rodents, short-term 3HAA treatment is beneficial in acute models of cardiovascular disease[21], spinal cord injury[22,23], asthma[24], and autoimmune encephalomyelitis[25]. Chronically elevating 3HAA during normal aging, analogous to our *C. elegans* experiments, has not previously been examined in mammals. We first conducted a small pilot study to evaluate the efficacy of 3HAA to impact aging in wild type mice. We fed

a small cohort of male C57BL/6J mice a diet supplemented with 0 (control, N = 8), 312.5 (low dose, N = 6), or 3125 (high dose, N = 7) ppm 3HAA starting at 27 months of age and measured survival and several metrics of health. Animals on both 3HAA diets were long-lived relative to mice on the control diet, with the low dose group living the longest (Fig. 4a). Neither diet had a significant impact on body weight (Fig. 4b). In parallel, we conducted a second small study to evaluate the impact of knocking out *Haao* (*Haao*[-/-]) on lifespan in male and female C57BL/6 N mice. *Haao*[-/-] animals (N = 24 females, 21 males) were significantly longer lived (Fig. 5a) and had modestly lower body weight during early- to mid-adulthood relative to wild type (N = 15 females, 22 males; Fig. 5b). The impact of *Haao* knockout on lifespan and body weight was driven by larger differences in female mice that reached statistical significance (Supplementary Fig. 9a, b), while male mice displayed smaller differences that did not reach significance (Supplementary Fig. 9c, d). The low and high dose 3HAA diets increased serum 3HAA concentration by ~3- and ~17-fold, respectively, and urine 3HAA concentrations by ~13- and ~200-fold, respectively (Fig. 4c). Similarly, *Haao*[-/-] mice had plasma and urine 3HAA concentrations that were ~90- and ~1000-fold higher than wild type, respectively (Fig. 5c). TRP, KYN, KA, and AA concentrations in serum and urine were similar in in mice fed control vs. 3HAA diets, and in *Haao*[-/-] vs. wild type mice, with the exception of KYN, which was elevated in the urine of mice fed control diet but not 3HAA diet after 10 weeks, (Supplementary Fig. 9e, f). Mice fed low dose 3HAA diet had a trend toward improved grip strength after 13 weeks (Fig. 4d) and *Haao*[-/-] mice had significantly higher grip strength than wild type at both 12 and 18 months of age (Fig. 5d). Low dose 3HAA diet significantly improved rotarod performance after 13 weeks; however, this may be a result of survivor bias, as individual improvement among mice surviving to 13 weeks was not significantly higher (Supplementary Fig. 9h).

Previous studies indicate that 3HAA is anti-inflammatory[21–25]. We observed several changes in immune cell populations that suggest a shift toward reduced inflammation in aging mice fed 3HAA diets. First, low dose 3HAA mice had more monocytes and fewer neutrophils and granulocytes within the myeloid lineage (Fig. 4e). Mice on both 3HAA diets had reduced percentages of nature killer (NK) cells within several sublineages (Fig. 4f) and resisted changes to T cell (Fig. 4g) and B cell (Fig. 4h) populations observed over 10 weeks, relative to control mice. Mice on both 3HAA diets also had reduced inflammatory and elevated resident monocytes (Supplementary Fig. 9i) but, like rotarod performance, changes in individual animals did not reach statistical significance. We did not observe notable differences in other parameters measured (Supplementary Fig. 9j–q).

## Discussion

Here we report that elevating the kynurenine pathway metabolite 3HAA through HAAO inhibition or 3HAA supplementation extends lifespan and improves late life motility in *C. elegans*. 3HAA increased resistance to oxidative stress while simultaneously inhibiting exogenous $H_2O_2$ production and increasing endogenous $H_2O_2$. Elevating 3HAA strongly activated the Nrf2/SKN-1 oxidative stress response and lifespan extension from both 3HAA and *haao-1(RNAi)* was largely abrogated in a genetic background expressing a hypomorphic allele of *skn-1*. Given the increase in endogenous $H_2O_2$, 3HAA likely activates Nrf2/SKN-1 canonically downstream of ROS generation. While the mechanism is similar in mammals and *C. elegans*, upstream signaling and the specific set of ubiquitination machinery (ubiquitin ligase and associated adapter proteins) are distinct[26] and the role of these proteins remains to be verified. We also have not rule out alternative or additional regulation via non-canonical activation of Nrf2/SKN-1[27]. These results implicate Nrf2/SKN-1 activation as one aspect of HAAO- and 3HAA-mediated health benefits. Importantly, lifespan extension from *haao-1* knockdown was not completely abrogated by *skn-1* loss-of-function, and also showed a complex and temperature-dependent

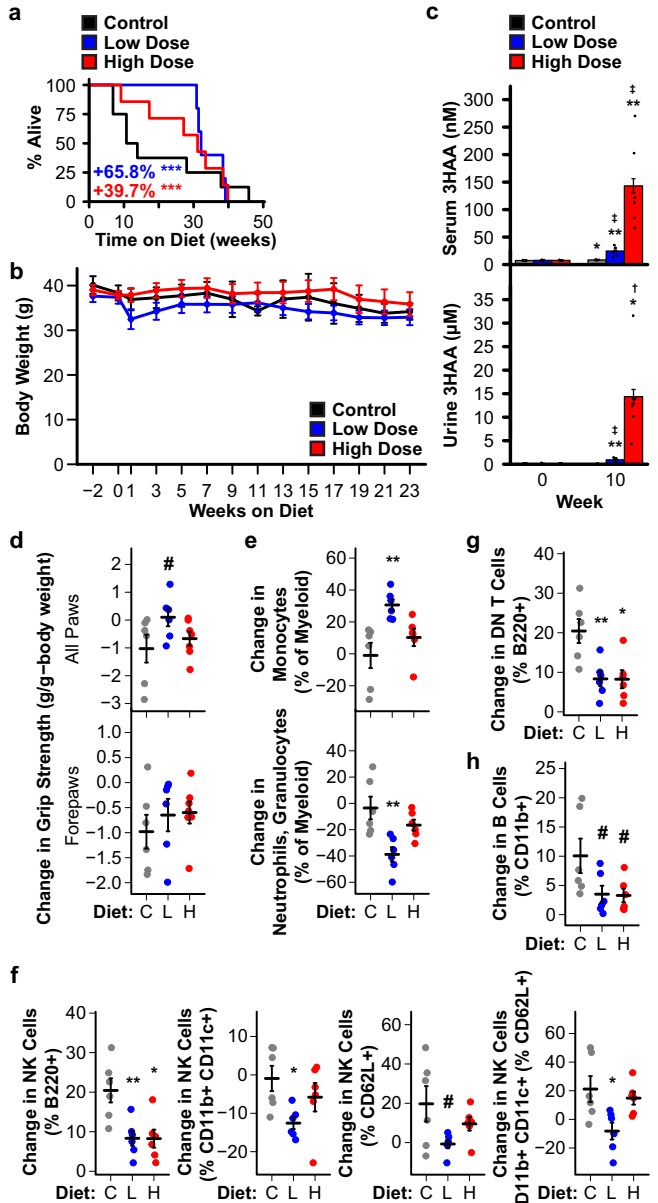

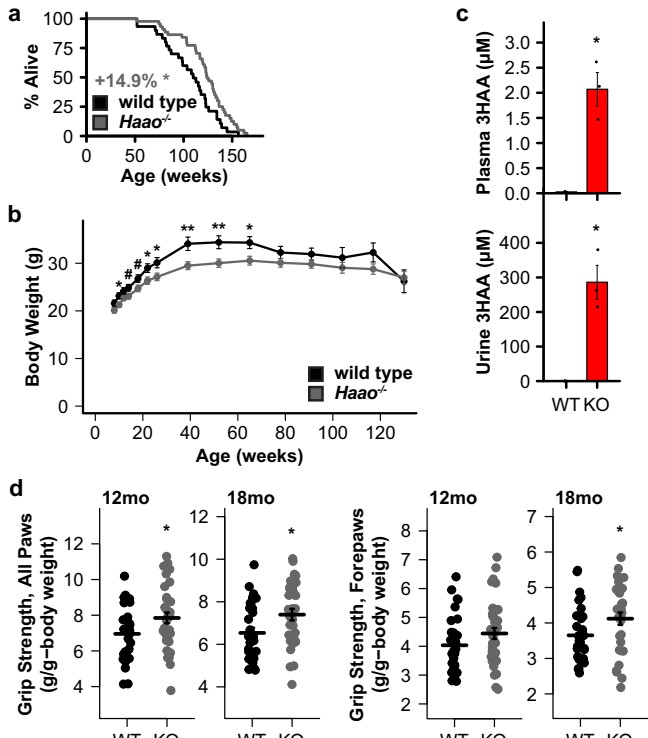

**Fig. 5 | Knocking out *Haao* extends lifespan and improves health metrics in mice. a** Deletion of Haao (*Haao⁻/⁻*, KO; *N* = 45) extends lifespan relative to wild type (*N* = 37) in C57BL/6N mice. Kaplan-Meier survival curves show lifespan from birth. Gray text indicates change in mean lifespan. **b** *Haao⁻/⁻* mice have lower body weight in early- to mid-adulthood. **c** *Haao* deletion increases 3HAA concentration in plasma and urine. **d** *Haao⁻/⁻* mice had significantly increased grip strength at 12 and 18 months of age. #$p < 0.1$, †, *$p < 0.05$, ‡, **$p < 0.01$, ***$p < 0.001$ vs. control diet (log rank test, panel **a**; two-sided Welch's *t* test, panels **b**–**d**). Error bars indicate standard error of mean. Summary statistics are provided in Supplementary Data 1 and source data are provided in Source Data.

**Fig. 4 | Dietary 3HAA extends lifespan, improves health metrics, and promotes a less inflammatory immune cell profile in mice. a** C57BL/6J mice fed chow supplemented with 312.5 ppm (low dose, L; *N* = 6) or 3125 ppm (high dose, H; *N* = 7) 3HAA starting at 27 months of age live longer than mice fed control diet (C; *N* = 8). Kaplan-Meier survival curves show remaining lifespan following introduction of the 3HAA diet. Colored text indicates change in mean remaining lifespan relative to control. **b** Neither 3HAA supplemented diet affected body weight. **c** Dietary 3HAA supplementation increases 3HAA concentration in serum and urine. **d** Mice on low dose 3HAA diet trended toward increased all paw grip strength after 13 weeks. Dietary 3HAA alters circulating immune cell profiles after 10 weeks: **e** shifting myeloid cells away from neutrophils/granulocytes and toward monocytes, **f** decreasing or preventing an increase in natural killer (NK) within several sub-lineages, **g** preventing an increase in double negative (DN) T cells, and **h** creating a trend toward reducing B cell counts. #$p < 0.1$, †, *$p < 0.05$, ‡, **$p < 0.01$, ***$p < 0.001$ vs. control diet (Wald test in Cox regression with weighted estimation, panel **a**; two-sided Welch's *t* test, panels **b**–**h**). For panel **d**, *, **, *** indicate significance vs. Week 0; †, ‡ indicate significance vs. control diet. Error bars indicates standard error of mean. Summary statistics are provided in Supplementary Data 1 and source data are provided in Source Data.

interaction with other aging pathways, suggest that a complete mechanistic model is multi-faceted. Finally, we present evidence that knocking out *Haao* throughout life or dietary 3HAA supplementation late in life can extend mouse lifespan. While these studies were small, they provide early promise, pending experimental replication, for elevating physiological 3HAA as a therapeutic target for lifespan extension and age-associated disease.

The kynurenine pathway is increasingly recognized as an important player in aging and age-associated disease. This study adds to a growing body of work on kynurenines in *C. elegans* aging by implicating *haao-1* as a fourth genetic target capable of influencing longevity that acts via mechanisms that are at least partially distinct from *tdo-2*, *kynu-1*, and *acsd-1*. Our data support 3HAA as the primary mediator of lifespan extension from *haao-1* knockdown. While not extensively tested, available data suggests regulatory feedback between kynurenine pathway enzymes and metabolites. In particular, our metabolite data indicate that *haao-1* knockdown results in a consistent increase in KYN and a decrease in AA, suggesting that elevating 3HAA may generate negative feedback on KYNU (Fig. 1a, Supplementary Fig. 6d, e, j, k). AA was consistently downregulated in response to knockdown of *tdo-2*, *kynu-1*, or *haao-1*, while supplementing worms with AA produced a dose-dependent decrease in lifespan. This suggests that reduced physiological AA may serve as a common mechanism between these three interventions. Additional work to understand the details of cross regulation between kynurenine pathway metabolites and enzymes and the concentration and mechanism of AA toxicity will be needed to validate and fully explore this model.

Even if is correct, 3HAA is still likely the primary mediator of lifespan extension from reduced *haao-1* because (1) among *tdo-2*, *kynu-1*, and *haaa-1*, knockdown of *haao-1* produces the largest lifespan extension and the least reduction in AA (Fig. 1b, Supplementary Fig. 6e, k), (2) knocking down *haao-1* in combination with either *kynu-1* or *tdo-2* tends to produces lifespan that resembles *kynu-1* or *tdo-2*, respectively (Supplementary Fig. 7b–d), and (3) 3HAA supplementation can further extend the already long lifespan of animals lacking *tdo-2*, *kynu-1*, or both *haao-1* and *kynu-1* (Fig. 3h, Supplementary Fig. 7a).

Our data demonstrate that 3HAA is protective against ROS in the context of aging *C. elegans*. 3HAA has a complex history with oxidative stress, with earlier studies linking 3HAA to ROS generation and/or oxidative damage[28–31] and more recent studies reporting antioxidant properties for 3HAA[32–39]. These studies vary widely in the oxidant properties examined, presence of added metals, cell type, and cellular context. 3HAA can auto-oxidize under specific conditions, including the presence of $Cu^{2+}$, $Fe^{3+}$ or alkaline pH[28]. Each of these variables is likely critical to understanding redox properties of 3HAA in the context of aging, and the particular importance of copper and pH were recently confirmed in silico[15]. Our data that 3HAA can both directly degrade $H_2O_2$ and promote endogenous $H_2O_2$ production suggest that both sides of this debate are likely correct in context. Given both that 3HAA is protective against oxidative stress and the complex interaction between *haao-1* and genes in other aging pathways (Fig. 1g), we speculate that 3HAA may also impart protection against other forms of cellular stress.

Downstream of the kynurenine pathway, accumulating evidence supports a model in which elevating $NAD^+$ is beneficial for healthy aging, including in *C. elegans*[40]. Knockdown of *tdo-2*, *kynu-1*, or *haao-1* all increase lifespan (Fig. 1b) while moderately decreasing $NAD^+$ in worms (Supplementary Fig. 1b). This implies that HAAO inhibition and 3HAA supplementation may impart benefits through different secondary mechanisms, with 3HAA acting as an $NAD^+$ precursor when active HAAO is present. If this is the case, synergistic benefits may abound when selected kynurenine pathway enzyme inhibitors are combined with strategies to increase $NAD^+$, such as supplementing $NAD^+$ precursors or inhibiting $NAD^+$-consuming enzymes. Given these complexities, understanding the interplay between kynurenine pathway enzymes and metabolites will be important to developing a comprehensive model for the role of kynurenine metabolism in aging.

In this work we examine the impact of chronic elevation of physiological 3HAA in whole animals during normal aging. Our data suggest that elevating 3HAA holds promise as a therapeutic objective in aging and age-associated disease; however, there are potential limitations to this approach that should be considered. First, our data suggest that inhibiting HAAO may slow growth early in life in both worms (Fig. 1c) and mice (Fig. 5b) and at least delay reproductive potential (Supplementary Fig. 1a). Thus, it may be important to only pursue therapeutic elevation of 3HAA as a late-life intervention. Second, historical data identified 3HAA as a possible mutagen and carcinogen. In vitro, one study found that 3HAA can promote chromosomal instability in human cell culture[41], while others observed that 3HAA did not induce mutagenesis in bacteria[42,43]. Similarly, several in vivo studies found that a solid pellet containing 3HAA implanted in the bladder increased the incidence of bladder cancer in mice and rats[44–46] while other studies did not observe 3HAA induced cancer via the same method[44,45,47]. More recent work suggests that 100 mg/kg 3HAA injected interperitoneally can inhibit hepatocellular carcinoma tumor growth in mouse xenograft models and enhance the anti-tumor impact of sorafenib, a first-line drug for treating HCC in humans[48–50]. The carcinogenic potential for 3HAA may simply be a matter of dose. The disparate results from the early bladder studies may be explained, in part, by the composition of the pellet.

Carcinomas were observed following implantation of a 3HAA-cholesterol pellet in some cases[44,45] (though not universally[47]), but not a 3HAA-paraffin pellet[44,45]. 3HAA is released from cholesterol pellets (~10% of 2 mg 3HAA released with 24 hours) at a substantially higher rate than from paraffin pellets (~6% of 2 mg 3HAA released over 10 days)[44]. Based on this diffusion data, and assuming a typical urine output volume for mice (~1 mL/day[51,52]), suggests an average urine 3HAA concentration around 1.2 mM for 3HAA-cholesterol pellets and around 0.2 mM for paraffin pellets within the first few days of implantation. While urine levels were substantially higher in our 3HAA fed and *Haao⁻/⁻* (Figs. 4c, 5c) mice than their respective controls, even the highest concentration measured was well below above estimates for the 3HAA-cholesterol pellets and similar to the estimate for the 3HAA-parffin pellets. This suggests that the urinary concentrations in our studies are well below observed carcinogenic levels, with the caveat that many experimental details that can impact these urine concentration estimates in the bladder experiments—animal age, pellet size, mouse strain—were missing from some or all of the early studies[41,44–46]. Schlegel et al.[46] observed that high dose ascorbic acid prevented tumor formation in mice with 3HAA-cholesterol pellets implanted in the bladder, suggesting that either an oxidation product of 3HAA, or ROS generated downstream of 3HAA, may mediate tumorigenesis. The molecular context in which 3HAA is tested, including media or environmental pH, the concentration of metal ions (e.g., copper[53]), or a direct interaction with cholesterol, are likely to be relevant factors in the potential carcinogenicity of 3HAA. We further examined the concentration of 3HAA needed to produce toxicity in cell culture for human embryonic kidney (HEK293) and mouse embryonic fibroblast (3T3) cells and found that 3HAA began to result in toxicity at concentrations at 800 μM or above (Supplementary Fig. 5f), which is well above even the urine 3HAA concentration in *Haao⁻/⁻* mice (Fig. 5c). The mouse studies presented here did not include tumor burden as an outcome, but we will examine this issue in detail in future studies. A third consideration is the inherent limitation to the mouse studies presented in this work. In the *Haao* knockout study, lifespan extension only reached statistical significance in females and the lifespan of wild type mice was short relative to other published studies (Supplementary Fig. 9a, c), so we cannot rule out the possibility that *Haao* knockout simply rescued the underlying cause of the reduced lifespan in this context. The 3HAA diet study included only male mice and the observed lifespan data violated the proportional hazard assumption (see description in Supplementary Methods). These limitations likely result from the fact that both experiments were designed as pilot studies and had small sample sizes. The observed differences between males and females in the *Haao* KO study may be a consequence of differential production of, or sensitivity to, 3HAA between males and females. Similarly, the fact that significant lifespan extension was observed in male mice in response to dietary 3HAA but not *Haao* knockout may reflect several differences between these interventions. First, the effective dose experience by the mice was likely different as the measured blood and urine concentrations were ~17- and ~24-fold higher for the *Haao* knockout mice relative to the mice fed even the high-dose 3HAA diet (Figs. 4c, 5c). Furthermore, the relative tissue exposure to 3HAA is likely quite different for 3HAA absorbed through the diet relative to 3HAA produced in *Haao* knockout mice in tissues with normally high HAAO activity. The dietary 3HAA intervention was also initiated late in life, while the *Haao* knockout mice experienced elevated endogenous 3HAA from birth. These issues motivate more detailed studies of 3HAA dose, intervention timing, and exploration of pharmacological HAAO inhibition as alternative strategies in future studies. Keeping this context in mind, the positive outcome in the mouse studies combined with the robust improvement in both lifespan and several metrics of heathy aging in *C. elegans*

provides a hopeful outlook for 3HAA as a therapeutic target in aging and motivates well-powered preclinical validation studies.

## Methods

### *C. elegans*

**Strains.** The following strains were obtained from the *Caenorhabditis* Genetic Center (CGC) in the College of Biological Sciences at the University of Minnesota: *daf-16(mu86) I* (CF1038), *eat-2(ad465) II* (DA465), *hif-1(ia4) V* (ZG31), *rsks-1(ok1255) III* (RB1206), *sir-2.1(ok434) IV* (VC199), *rrf-1(pk1417) I* (MAH23)[54], *rde-1(ne219) V; kzIs9[(pKK1260) lin-26p::NLS::GFP + (pKK1253) lin-26p::rde-1 + rol-6(su1006)]* (NR222), *lin-15B(n744) X; uIs57[unc-119p::YFP + unc-119p::sid-1 + mec-6p::mec-6]* (TU3335), *rde-1(ne219) V; kbIs7[nhx-2p::rde-1 + rol-6(su1006)]* (VP303), *rde-1(ne300) V; neIs9[myo-3::HA::RDE-1 + rol-6(su1006)] X* (WM118), *wgIs341[skn-1::TY1::EGFP::3xFLAG + unc-119(+)]* (OP341), *dvIs19[(pAF15) gst-4p::GFP::NLS]* (CL2166), *ldIs3[gcs-1p::GFP + rol-6(su1006)]* (LD1171)[55], *muIs84[(pAD76) sod-3p::GFP + rol-6(su1006)]* (CF1553), *skn-1(zj15) IV* (QV225), and *jrIs1[rpl-17p::HyPer + unc-119(+)]* (JV1)[18]. Strains *kynu-1(tm4924) X* (FX04924; backcrossed 6x to N2 to create strain GLS129) and *haao-1(tm4627) V* (FX04627; backcrossed 6x to N2 to create strain GLS130) were obtained from the *C. elegans* National Bioresource Project (NBRP) at the School of Medicine at the Tokyo Women's Medical University. Wild type (N2) worms, *Escherichia coli* OP50 bacteria, and *E. coli* HT115 empty vector (EV) bacteria were originally obtained from Dr. Matt Kaeberlein (University of Washington, Seattle, WA, USA). Strain RB1206 was generated by the *C. elegans* Gene Knockout Project at the Oklahoma Medical Research Foundation as part of the International *C. elegans* Gene Knockout Consortium[56]. Strain VC199 was generated by the *C. elegans* Reverse Genetics Core Facility at the University of British Columbia as part of the International *C. elegans* Gene Knockout Consortium[56]. Strain OP341 was constructed as part of the Regulatory Element Project, part of modENCODE[57]. Strain TU3335 was created by Calixto et al.[58]. Strain VP303 was created by Espelt et al.[59]. Strain LD1171 was created by Wang et al.[55]. Strain *haao-1(tm4627) V kynu-1(tm4924) X* (GLS147) was generated by crossing GLS129 to GLS130. Strains *haao-1(syb2665)::wrmscarlet* (PHX2665) and *kynu-1(syb2691)::wrmscarlet* (PHX2691) were created for this work using CRISPR/Cas9 precise gene insertion by SunyBiotech Corporation. RNAi feeding vectors in the *E. coli* HT115 background were obtained from the Ahringer[60,61] (Source Bioscience) or Vidal[62] (Horizon Discovery Darmacon Reagents RCE1181) RNAi feeding libraries. Additional strain information is provided in Supplementary Data 1 (Supplementary Tables 22, 23).

**Media and culture.** We maintained hermaphrodite worms on solid nematode growth media (NGM) seeded with *Escherichia coli* OP50 bacteria at 20 °C as previously described[63] except where otherwise noted. RNAi experiments used *E. coli* strain HT115, while all other experiments used strain OP50. All experiments were conducted using hermaphrodites. We conducted RNAi feeding, lifespan, healthspan, brood size, and fluorescence quantification assays according to standard protocols. All worms were transferred to NGM plates containing 50 µM 5-fluorodeoxyuridine (FUdR) starting at the L4 larval stage to prevent reproduction.

**RNA interference (RNAi).** All experiments were conducted on NGM containing 1 mM Isopropyl β-D-1-thiogalactopyranoside (IPTG) to activate production of RNAi transcripts and 25 µg/mL carbenicillin to select RNAi plasmids and seeded with live *E. coli* (HT115) containing RNAi feeding plasmids. Worms were age-synchronized via timed egg laying (with the exception of oxidative stress assays, which were synchronized via bleach prep) at the experimental temperature and transferred to plates containing 50 µM 5-fluorodeoxyuridine (FUdR) to prevent reproduction at the L4 larval stage as previously described[63]. We validated both *haao-1(RNAi)* and *kynu-1(RNAi)* by confirming that fluorescence in HAAO-1::wrmscarlet and KYNU 1::wrmscarlet

transgenic strains were reduced below detectable levels (Supplementary Fig. 5b). Additional information on RNAi clones is provided in Supplementary Data 1 (Supplementary Table 23).

**3HAA supplementation.** 3HAA supplementation was achieved by adding the appropriate mass of solid 97% pure 3HAA (Sigma-Aldrich, catalog number 148776) to NGM media during preparation prior to autoclaving.

**Lifespan analysis.** Lifespan and paralysis experiments were conducted as previously described[63]. Briefly, adult animals were maintained on NGM RNAi plates with FUdR throughout life. Each animal was examined every 1–2 days (25°) or 2–3 days (15 °C, 20°) by nose- and tail-prodding with a platinum wire pick. Animals were scored as dead if they failed to react to prodding. Live and dead animals were counted, and dead animals removed from the plate at each examination. Animals displaying vulva rupture were included in all analyses, while worms that left the surface of the plate were excluded. Technicians scoring each experiment were blinded to the experimental group identity until after all worms were scored as dead.

Each lifespan experiment included test groups consisting of 105–150 worms (3 plates with 35–50 worms/plate). Each experiment included a negative control test group fed the *E. coli* strain matched to the test conditions. For lifespan epistasis and tissue-specific RNAi experiments, each combination of candidate gene RNAi and mutant/transgenic strain was measured in three independent experiments, each including: wild type worms fed *EV(RNAi)*, wild-type worms fed candidate RNAi, mutant worms fed *EV(RNAi)*, and mutant worms fed candidate RNAi. P-values for statistical comparison of lifespan between test groups were calculated using the log rank test (*survdiff* function in the R version 4.3.0 "survival" package) with Holm multiple test correction applied to comparisons made within each experiment. Summary statistics for all lifespan experiments are provided in Supplementary Data 1 (Supplementary Tables 1, 5, 6, 9, 10, 13, 14, 17, 21).

### Kynurenine pathway metabolite quantification (The Jackson Laboratory)

**Metabolite extraction.** Worms were grown plates containing live *E. coli* (HT115) containing RNAi feeding plasmids targeting *kynu-1(RNAi)*, *haao-1(RNAi)*, or *tdo-2(RNAi)* at 15 °C under conditions identical to those used in the lifespan studies. On day 4 of adulthood, ~100 worms/sample were collected from the plates and washed twice with M9 buffer (21.6 mM $Na_2HPO_4$ 22 mM $KH_2PO_4$ 85.6 mM NaCl, 1 mM $MgSO_4$). Excess M9 was removed and the worm pellets flash frozen in liquid nitrogen and stored at −80 °C prior to metabolite extraction. To extract metabolites, 1 ml of extraction buffer (2:2:1 AcN:MeOH:H2O + internal standards [0.2 ng/ul 1-Napthylamine + 0.2 ng/ul 9-anthracene carboxylic acid]) was added to each sample. Samples were homogenized mechanically and sonicated to dissolve worm pellet and placed at −20 °C overnight to precipitate metabolites. Samples were then centrifuged at 20,000 × *g* for 15 min to pellet protein, and the supernatant transferred to a new tube and dried without heat. Metabolites were resuspended in a solution of 10% acetonitrile in water, both Optima® grade for mass spectrometry analysis. Each sample was collected in biological triplicate.

**Relative quantification via mass spectrometry (MS).** Samples were run on an Agilent 6530 Q-TOF equipped with a microflow liquid chromatography (LC) system. Separation by reverse-phase LC (RPLC) with a C18 column (Agilent Poroshell, 2.1 mm × 50 mm) over a 20 min gradient resulted in separate peaks for each analyte. Detection was in positive ion mode over a mass range of 200–1700 *m/z* with internal reference ions. Collision-induced dissociation (CID) was employed to confirm the identification of each analyte by mass and retention time. For preliminary analysis, both automated detection and fragmentation

of all analytes, as well as detection and fragmentation of targeted analytes were employed. Both resulted in sufficient fragmentation for identification and sufficient intensity for quantification. Subsequent analysis employed quantitative scans for confirmation of relative abundance.

**Data analysis.** Samples were extracted for target ions using ACD Labs MSWorkbookSuite IntelliTarget feature (http://www.acdlabs.com/products/spectrus/workbooks/ms/msworkbooksuite/). Briefly, a database for the targets of interest was generated and populated using standard compounds as well as library reference data. Targets were extracted and quantified with normalization against an internal standard. Scans were verified manually at both the first and second stages of MS (MS1 and MS2).

We gratefully acknowledge the contribution of Dorothy Ahlf Wheatcraft in the Protein Sciences Service at The Jackson Laboratory for expert assistance with this portion of the work. This methodology was used in the experiments reported in Fig. 1c and Supplementary Fig. 6a–f. Summary statistics for these experiments are provided in Supplementary Data 1 (Supplementary Table 7).

### Kynurenine pathway metabolite quantification (University of Arizona)

**Metabolite extraction.** For *C. elegans* metabolite quantification, ~1000 animals were collected from NGM plates and washed 3 times in M9. All M9 was carefully removed and the worm pellet flash frozen in liquid nitrogen. For metabolite extraction, worm pellets were thawed and 100 μL of PBS added, followed by 200 μL acetonitrile. Samples were vortexed, followed by sonication in a water bath sonicator for 15 min. To 50 μL of the prepared worm sample, 10 μL of internal standard is added followed by 40 μL of 0.2% formic acid in acetonitrile and vortexed. Samples were centrifuged at $15,800 \times g$ at 4 °C for 5 min and the supernatant transferred to autosampler vials and 1 μL injected for analysis.

**UPLC-MS.** Analysis of kynurenine metabolites was performed on an Ultivo triple quadrupole mass spectrometer combined with a 1290 Infinity II UPLC system (Agilent, Palo Alto, CA). The instrument was operated in electrospray positive mode with a gas temperature of 180 °C at a flow of 7 L/min, nebulizer at 35 psi, capillary voltage of 3000 V, sheath gas at 275 °C with a flow of 8 L/min and nozzle voltage of 300 V. The instrument was operated in dynamic multiple reaction monitoring mode (dMRM) with positive ionization. Chromatographic separation was achieved using a flow rate of 0.4 mL/minute and a gradient system of water (A) and acetonitrile (B), both with 0.2% formic acid (v/v) on an Acquity UPLC BEH C-18 1.7 u 2.1 × 100 mm column (Waters, Milford, MA) maintained at 40 °C. After each injection, the column was re-equilibrated for 5 minutes prior to the next injection. Samples were maintained at 4 °C.

**Calibration.** Mixed calibration solutions were prepared by serial dilution of a mixed stock solution containing all analytes of interest. Mixed internal standard solutions were prepared by serial dilution of a stock solution containing all internal standards. Calibration solutions and internal standard solutions were diluted in water. Nine-point calibration curves ranging from approximately 2000 to 0.1 pmole/sample were prepared for each analysis by adding 10 μl internal standard solution to 10 μl standard solution. Prior to analysis, 40 μL of 0.2% formic acid in acetonitrile was added to the calibration samples, which were then vortexed and centrifuged at $15,800 \times g$ at 4 °C for 5 min. Supernatant was transferred to autosampler vials and 1 μL injected for analysis. Linear regression was performed by plotting the peak area ratio of analyte to internal standard vs. concentration and weighting the regression curve 1/x. Data analysis was conducted using Agilent Mass Hunter software. We gratefully acknowledge the contribution of

Sherry Chow and Wade Chew in the Protein Sciences Service at University of Arizona Cancer Center (UACC) Analytical Chemistry Shared Resource (ACSR) for expert assistance with this portion of the work. This methodology was used in the experiments reported in Supplementary Fig. 6g–k. Summary statistics for these experiments are provided in Supplementary Data 1 (Supplementary Table 8).

**Brood size.** To measure brood size, 10 worms at the L4 larval stage were placed on individual NGM RNAi plates lacking FUdR and allowed to lay eggs. Worms were transferred to a new plate every 12 h at 25 °C or every 24 h at 15 °C or 20 °C until worms ceased laying eggs. The number of progeny was counted on each plate following a 2 day incubation to allow eggs to hatch. Welch's *t*-tests were used to determine significance in observed differences between target and EV RNAi at each age, and between total progeny counts. Summary statistics for brood size experiments are provided in Supplementary Data 1 (Supplementary Tables 4, 11).

**Health metrics.** Health data in Fig. 1c, d was collected in three independent experiments using the WormLab system (Version 3.1, MBF Bioscience, Williston, VT) at 7, 14, 21, and 25 days of age (15 °C) and 4, 8, 11, and 15 days of age (25 °C). For crawling speed assays, videos were captured for worms directly on NGM experiment plates. Worms were motivated to begin moving by dropping the plate onto the WormLab video capture stage from a height of ~0.5 inches. Video capture was started immediately and allowed to run for 1 min, 15 s. Worms reacted to the plate drop by moving rapidly for approximately 20 s then returning to a normal foraging behavior. For "motivated speed", worms were tracked using the WormLab software for the first 15 s of each video. Worms present for at least 14 s were selected for analysis. For "unmotivated speed", worms were tracked between 30 s and 1 min, 15 s. Worms present for at least 30 s were selected for analysis. For thrashing, worms were suspended in a droplet of M9 buffer on unseeded NGM plates and video captured for 45 s. Worms present for at least 15 s were selected for analysis. Average speed, thrash frequency, body length, and body width for each worm was exported from the WormLab software to R version 4.3.0 for analysis. The WormLab software tended to misidentify dark sections of the plate as worms. To filter these errors, we initially removed any identified object that fell outside of 1.2 standard deviations in worm body length, width, or area. Based on manual inspection, this threshold removed non-worm objects and poorly identified worms without removing correctly identified worms. Thrashing data in Fig. 2g was collected manually by suspending worms in a droplet of M9 buffer on unseeded NGM plates and tallying thrashes for 6 s and entering data in R version 4.3.0 for analysis. Two-sided Welch's *t*-tests were used to determine significance in differences in each phenotype between target and EV RNAi at each age. Summary statistics for these experiments are provided in Supplementary Data 1 (Supplementary Tables 2, 3, 12).

**Red pigmentation imaging.** Images of wild type and *haao-1(tm4627) C. elegans* to illustrate accumulation of red pigment were taken using a Samsung S9+ Smartphone using stock camera software operating in professional mode attached to a Zeiss Stemi 508 stereo dissection microscope using a 3d-printed cell phone mount. All camera settings were constant across images.

**Paraquat toxicity assay.** To assess paraquat toxicity, survival assays were conducted as described above with worms maintained on NGM containing 20 mM paraquat starting at L4. All animals were scored daily for lifespan.

**Hydrogen peroxide ($H_2O_2$) measurement.** $H_2O_2$ was quantified using Invitrogen™ Amplex™ UltraRed Reagent (ThermoFisher Scientific catalog number A36006) according to manufacturer instructions,

including use of the optional Invitrogen™ Amplex™ Red/UltraRed Stop Reagent (ThermoFisher Scientific catalog number A33855). For in vitro $H_2O_2$ degradation assays, stock solutions of 3HAA and $H_2O_2$ in water were mixed to achieve the indicated concentrations and incubated at room temperature for 30 min prior to quantification. For in vivo $H_2O_2$ in *C. elegans*, age-synchronized populations were grown as described above. 100 worms/sample were collected on day 6, 13, and 20 from egg with 5 technical replicates per test condition. Worms were collected and washed 2 times with M9 0.05% Triton X-100, and incubated at 20 °C with shaking in 1x reaction buffer (50 mM $NaH_2Po_4$, 0.05% Triton X-100 pH 7.4) for 3 h. After incubation, the worms were allowed to settle to the bottom of the tube (3–5 min), after which the supernatant was collected for quantification.

**Fluorescence microscopy.** To quantify fluorescence in endogenous $H_2O_2$ (Hyper), *haao-1::mScarlet*, *kynu-1::mScarlet*, *SKN-1::GFP*, *gst-4p::GFP*, *gcs-1p::GFP*, and *sod-3p::GFP* transgenic strains, age-synchronized populations of *C. elegans* were maintained as described above. At the designated ages, 1–20 animals were manually transferred to a drop of 25 mM levamisole on slides prepared with 6% agarose pads and imaged using a Leica M205 FCA Fluorescent Stereo Microscope equipped with a Leica K6 sCMOS monochrome camera. Identical imaging settings were maintained across timepoints for each strain. Fluorescence was quantified using LightSaver (version 0.1)[64]. Each experiment was repeated in biological triplicate. Analysis was conducted in GraphPad Prism (version 8. 4.3.686). Summary statistics for these experiments are provided in Supplementary Data 1 (Supplementary Tables 15, 16).

**Mice**

**Lifespan studies**

**3HAA diet study.** Male C57BL/6J mice (stock # 000664) were bred and aged at The Jackson Laboratory to 27 months while being housed in a pathogen-free room with a 12:12 h light:dark cycle and given a standard rodent diet (LabDiet 5KOG) and acidified water. At 27 months, animals were randomly assigned by cage to diets with either 0 ppm 3HAA (control; *n* = 8), a diet with 312.5 ppm 3HAA (low dose; Sigma-Aldrich, catalog number 148776; *n* = 6), or a diet with 3,125 ppm 3HAA (high dose; *n* = 7). This study was designed as a pilot study and only a limited number of mice were available. Sample size was not predetermined and all available mice were used. Control, low dose, and high dose diets were all prepared identically by LabDiet by grinding LabDiet 5LG6-JL, adding the appropriate mass of 3HAA, reforming pellets, and sterilizing by exposure to ionizing radiation. Blood and urine were collected at baseline (prior to starting assigned diet) and 10 weeks after starting the diet for blood chemistry, urine chemistry, and to determine blood cell composition using flow cytometry. Mice were kept until their natural death or until humanely euthanized by carbon dioxide inhalation because they reached a moribund state. All animal experiments were performed in accordance with the National Institutes of Health Guide for the Care and Use of Laboratory Animals (National Research Council) and were approved by The Jackson Laboratory's Animal Care and Use Committee. Only male mice were included in this study because of limited availability of aged mice.

**Haao knockout study.** Cryopreserved sperm from C57BL/6 N mice homozygous the *Haao^tm1b(KOMP)Mbp*^ mutation was obtained from the International Mouse Phenotyping Consortium (www.mousephenotype.org)[65] and used to recover live mice by the University of Arizona Cancer Center (UACC) Genetically Engineered Mouse Models (GEMM) to establish a breeding colony. Deletion of functional HAAO was confirmed by PCR, sanger sequencing, and accumulation of 3HAA in serum, plasma, or urine. Breeders were maintained by crossing male and female mice heterozygous for the *Haao^tm1b(KOMP)Mbp*^ allele (*Haao^+/-^*). Experimental mice were generated by

crossing male and female *Haao^+/-^* mice and selecting wild type (WT, *Haao^+/+^*; $N_{female} = 15$, $N_{male} = 22$) and homozygous (KO, *Haao^-/-^*; $N_{female} = 24$, $N_{male} = 21$) offspring and aged at the University of Arizona while housed in a pathogen-free room with a 12:12 hour light:dark cycle and given a standard rodent diet (NIH-31, Teklad 7913) and acidified water. The following primers were used for PCR and sanger sequencing: haao-f-ext (CTGAGCAGACAACACCAAGTAGCAACC), haao-f-int (CAGTATCGGCGGAATTCCAGCTGAG), and haao-r-ext (GGTGACAGTGGACATAGAACCTCCC). Primers haao-f-ext and haao-r-ext flank the deletion region and produce distinct PCR bands, while haao-f-int complements a region distinct to the *Haao^tm1b(KOMP)Mbp*^ allele and produces a band only in heterozygous or homozygous knockout mice. This study was designed as a pilot study. Sample size was not selected to optimize power; however, we added mice to a minimum of 15 mice/sex/group, which corresponds to 75% power to detect a 20% change in lifespan in the combined group, and 50% power to detect a 20% change in lifespan in each sex. Mice were kept until their natural death or until humanely euthanized by carbon dioxide inhalation because they reached a moribund state. All animal experiments were performed in accordance with the National Institutes of Health Guide for the Care and Use of Laboratory Animals (National Research Council) and were approved by The University of Arizona Animal Care and Use Committee.

**Statistics.** To evaluate statistical significance in mouse lifespan, we first constructed a Cox proportional hazard regression model from survival data (*coxph* function in R version 4.3.0 "survival" package) and assessed the proportional hazard assumption. If this assumption was satisfied ($p \geq 0.05$ using *cox.zph* function in the R version 4.3.0 "survival"), as was the case for the *Haao* knockout study we proceeded to assess statistical differences using a log rank test; if the assumption was not satisfied ($p < 0.05$ using *cox.zph* function in the R version 4.3.0 "survival"), which was the case for the 3HAA diet study, apparently driven by a single long-lived outlier in the control mouse group (*ggcoxdiagnostics* function in the R version 4.3.0 "survminer" package), we instead employed weighted estimate in Cox regression (*coxphw* function in the R version 4.3.0 "coxphw" package), a method robust to violations of the proportional hazard assumption[66]. *P*-values in the latter case reflect the results of a Wald test performed by the *coxphw* function on pair-wise comparisons of diet groups using this approach. Per a request during peer review, a post hoc analysis of the survival data was conducted using the log rank test (*survdiff* function in the R version 4.3.0 "survival" package) with Holm multiple test correction applied to correct for all four comparisons. The p values are reported in Supplementary Data 1. However, because the data violate the proportional hazard assumption, we do not consider the log rank *p* values to be interpretable. This methodology was used in the experiments reported in Figs. 4a, 5a, and Supplementary Fig. 9a, c. Summary statistics for these experiments are provided in Supplementary Data 1 (Supplementary Table 18).

**Health and behavioral assessment.** Grip strength and latency of fall from a rotarod were measured at both at baseline (27 months) and 30 months (3HAA diet study), or at 12 and 18 months of age (*Haao* knockout study) as previously described[67] (Ackert-Bicknell, 2015). Deficit accumulation frailty index (FI) is a metric of general health and resilience that incorporates assessment of 30 physical traits into a single index. FI was measured for all surviving mice at baseline (27 months) and every 2 weeks until a final assessment at 30 months (3HAA diet study), or at 12 and 18 months of age (*Haao* knockout study) as previously described[68]. Body weight measurement was collected as part of FI assessment in both studies, and every 2 weeks for the first 6 months and every 3 months thereafter for the *Haao* knockout study. The Y-maze assay to quantify number of arm entries and spontaneous alternations was performed for the 3HAA diet study

 

as previously described[68]. We gratefully acknowledge the use of facilities and equipment through the Center for Biometric Analysis at The Jackson Laboratory in this portion of the work for the 3HAA diet study.

**Glucose tolerance test.** Food access was removed for all mice 4 h prior to glucose tolerance test. Glucose, 2 mg/g body weight (G8769; Sigma-Aldrich, St. Louis, MO) was injected into the intraperitoneal cavity. Blood glucose was measured immediately prior to glucose injection (0 min) and 15, 30, 60, 120, and 180 min post-injection from blood from the tail vein using a NovaStatStrip Xpress handheld glucometer (Nova Biomedical, Waltham, MA). The 180-minute time point was excluded for the *Haao* knockout study.

**Clinical chemistry.** Triglycerides, cholesterol, and blood urea nitrogen (BUN) concentrations were measured in serum separated from whole blood using a Beckman AU680 Chemistry Analyzer, as were urine albumin and creatinine concentrations. Actual mouse albumin concentrations were calculated by linear regression from a standard curve generated with mouse albumin standards (Kamiya Biomedical Company, Seattle, WA)[67]. We gratefully acknowledge the contribution of Todd Hoffert in the Histopathology Service at The Jackson Laboratory for expert assistance with this portion of the work.

**Kynurenine pathway metabolite quantification (University of Arizona)**
**Metabolite extraction.** Plasma or urine samples (20 µL) were thawed and prepared by adding 10 µL internal standard solution and 50 µL 0.2% formic acid in acetonitrile to precipitate plasma proteins. Following vortex mixing, samples were centrifuged at $15,800 \times g$ at 4 °C for 5 min, the supernatant transferred to autosampler vials, and 1 µL injected for analysis.

**UPLC-MS, calibration, and analysis.** UPLC-MS, calibration, and analysis were performed as described above for *C. elegans* samples. This methodology was used in the experiments reported in Figs. 4c, 5c and Supplementary Fig. 9e, f. Summary statistics for these experiments are provided in Supplementary Data 1 (Supplementary Tables 19, 20).

**Flow cytometry.** Blood composition analysis was collected and analyzed by flow cytometry following a standardized protocol for mouse aging studies. This protocol has been previously described in extensive detail including sample collection and preparation, cell markers and fluorochromes targeted, instrumentation, software, and gating strategy[67]. We gratefully acknowledge the contribution of Will Schott in the Flow Cytometry Service at The Jackson Laboratory for expert assistance with this portion of the work.

**Body composition.** Whole body composition (fat and lean tissues) was measured in live, conscious mice by nuclear magnetic resonance using an Echo MRI Analyzer (Echo Medical Systems, Houston, TX).

**Western blots.** Cells were washed twice with PBS, lysed in a radio-immunoprecipitation analysis (RIPA) solution with 1 mM PMSF, sonicated for 30 seconds and incubated for 5 minutes at 95 °C. Protein concentrations were determined using the bicinchoninic acid (BCA) protein assay kit (Thermo Fisher Scientific, Cat. No. 23227) and equal amounts of protein by weight were combined with 6x Laemmli buffer, separated on 4–20% polyacrylamide gradient gels (Bio-Rad Laboratories) and transferred onto PVDF membranes (Bio-Rad Laboratories) after being activated for 30 s in methanol. Membranes were washed in PBS with 0.1% Tween-20 in PBS (PBST) and blocked in PBST containing 5% (w/v) milk for 1 hour. Membranes were incubated overnight in PBST containing 5% (w/v) milk and primary antibodies (1:000 dilution). After washing with PBST, membranes were incubated at room temperature for 1 hour in PBST containing 5% (w/v) dried milk and secondary

antibodies (1:3000 dilution). Immunoblots were imaged on an Azure 400 Imaging System (Azure Biosystems). The following antibodies were used: anti-Nrf2 (primary; Abcam ab62352, clone number EP1808Y, knockout validated by manufacturer in human cells), anti-β-actin (primary; Cell Signaling Technology 3700 S), goat anti-mouse IgG2b heavy chain (HRP) preabsorbed (secondary; Abcam ab98703, polyclonal), and goat anti-rabbit IgG2b heavy chain (HRP) preabsorbed (secondary; Santa Cruz Biotechnology sc-2004, polyclonal).

**Statistics.** Except where otherwise noted, two-tailed Welch's *t* tests were used to make comparisons between groups.

**Reporting summary**
Further information on research design is available in the Nature Portfolio Reporting Summary linked to this article.

## Data availability
The datasets generated and/or analyzed during the current study are available in Supplementary Data 1 and Source Data. Supplementary Information is available for this paper. Source data are provided with this paper.

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

## Acknowledgements

We thank Dr. Aric Rogers, Dr. Jarod Rollins, and Santina Snow at the Mount Desert Island Biological Laboratory (MDIBL) for helpful discussion. We further thank Dr. Rogers for use of the WormLab system. This work was supported by NIH P30AG038070 to Gary Churchill and RK, NIH R35GM133588 to G.L.S., P30CA023074 to Joann B. Sweasy, and the State of Arizona Technology and Research Initiative Fund administered by the Arizona Board of Regents. The mouse work was supported by a pilot award to G.L.S. from The Jackson Laboratory Nathan Shock Center for Excellence in Basic Biology of Aging (NIH P30AG038070). G.L.S. was supported as a Jackson Laboratory Scholar in Aging Award.

## Author contributions

H.D., R.C.P., and L.E. conducted primary experimental planning, execution, and analysis at the University of Arizona related to *C. elegans* lifespan measurement, microscopy, fluorescence quantification, and other molecular assays, and drafted the manuscript. SF developed software tools and conducted fluorescence quantification analyses. G.B., E.S., J.M., K.S., E.A.G. L.M.C., J.B., N.T., C.C., T.L., and S.B. collected and analyzed *C. elegans* lifespan, healthspan, reproduction, and fluorescence quantification data. S.S. provided training and experimental support for *C. elegans* and mouse work conducted at The Jackson Laboratory. R.K. supervised all aspects of the *C. elegans* and mouse lifespan studies carried out at The Jackson Laboratory. G.L.S. collected and analyzed *C. elegans* lifespan, healthspan, and reproduction data, supervised all aspects of the work carried out at the University of Arizona, and drafted the manuscript. All authors edited and approved the manuscript.

## Competing interests

G.L.S., S.F., and E.A.G. are co-founders, owners, and Managing Members of Senfina Biosystems, LLC. All other authors declare that they have no competing interests.
