## [Peer Review File · Nature Communications]

On the benefits of the tryptophan metabolite 3-hydroxyanthranilic acid in *Caenorhabditis elegans* and mouse agingREVIEWER COMMENTS

Reviewer #1 (Remarks to the Author):

The work “3-hydroxyanthranilic acid– a new metabolite for healthy lifespan extension” authored by Dang H. et al present data supporting a role for 3-hydroxyanthranilic acid (3HAA) in lifespan extension in *C. elegans*. Authors suggest the anti-ageing effect of 3-HAA is achieved due to the antioxidant properties of 3-HAA -both directly by scavenging H₂O₂ and indirectly by activating SKN-1 (NRF2 homolog). An experiment in mice is provided aiming at generalizing the finding reported in *C. elegans*. Overall the manuscript is well written and further explores a metabolic pathway that has been related to lifespan extension in previous reports. The authors present convincing data supporting the effect of 3-HAA on lifespan extension. On the other hand, their claim that 3HAA is “healthy” might contradict previous reports. I have also found not very convincing the mechanism(s) by which 3-HAA is driving life extension. The following issues should be addressed to clarify the mechanism and the reactivity of 3HAA:

Major issues.

1- 3-HAA has been previously reported to significantly impair tumor growth (doi: 10.1038/s41420-021-00561-6), and it was originally described as an endogenous mutagen (<https://doi.org/10.1038/177837a0>). The translational application of 3HAA might be limited considering the mutagenic and cytotoxic effect of 3HAA reported on mammalian cells.

Therefore, how authors reconcile their proposal for of 3HAA as therapeutic intervention? Is 3HAA a mutagen in your setting? They might perform toxicity and/or mutagenicity assays in cells to address this concern.

2- In Fig. 1F the results show similar overall life change in *kynu-1* and *haao-1*. If accumulation of 3HAA is driving the phenotypes, then 3HAA accumulation in HAAO-1 siRNA might be causing product inhibition of *kynu-1*, thus leading to phenotype. The Fig. 2F aims at resolving the epistatic interaction among *kynu-1*, *haao-1* in presence of 3-HAA. The control should be shown in each plot, helping the comparison. Authors should also perform/show ageing experiments to compare single and double knock-downs in absence of 3-HAA (like Fig. 1B but with double knockdown/out).

3- From the data presented in Fig. 3, it looks like 3HAA-anti-aging effect is more pronounced in absence of kynu-1 than in absence of haao-1. Hence, 3HAA might somehow enhance the effect of kynu-1 deletion, or as proposed, work through a completely different mechanism. Expanding the discussion of the role of kynu-1 would help to clarify the possible mechanisms of 3-HAA anti-aging. In addition, a clear additive effect in anti-ageing would be expected by growing worms with TRP and 3HAA -further supporting independent effects.

4- Authors claim the 3HAA effect might be independent of KYN. 1) they claim 3-HAA is redox active and induces the degradation of H₂O₂. A more in vivo demonstration of this effect would be to use the HyPer worms (see <https://doi.org/10.1038/s41467-022-28242-7>).

5- Authors show that haao-1 inactivation upregulates SKN-1 (NRF2) in worms. SKN-1 is usually activated by compounds that cause oxidative stress (see <https://doi.org/10.1038/s41467-022-28242-7>), thus it would be expected that the accumulation of 3-HAA in haao-1 knockdown could lead to repression of SKN-1 instead of activation. Does 3HAA show additive effect with other compounds able to modulate SKN-1?

6- Related to the previous issue, 7- SKN-1- mammalian ortholog (NRF2) is under the control of KEAP1. In worms, KEAP1-ortholog has not been described, and instead the oxidative stress response is triggered through a signaling cascade that results in the phosphorylation and activation of the MAPKK and P38/MAPK homologs. Are those signalling kinases also phosphorylated in response to 3HAA or in haao-1 mutant? Could authors speculate how SKN-1 would be activated by 3HAA in worms?

7- A simple test supporting a general role of 3HAA in regulating SKN-1 would be to measure NRF2-target genes in a mammalian cell in presence of 3HAA.

8-The epistatic analysis of SKN-1 haao-1 (Fig. 3g) does not contribute to clarify the relationship between skn-1 and haao-1 because it only said that the effect of skn-1 inactivation is dominant. A better experiment would be to show the lifespan but adding 3-HAA. This experiment would indicate if 3-haa effect is independent of skn-1 or necessitates of skn-1 (like Extended Fig. 6 with SKN-1 haao-1 strain).

Minor issues

1 -Fig. 1B, the legend is grey and the line black

2- Line 140, should be Fig. 2F

3- Extended Data Fig. 5. The phrase "3HAA is upregulated" is not appropriate. May be use "3HAA accumulation" instead.

Reviewer #2 (Remarks to the Author):

Based on previous published work from the same group, this manuscript supports the relevance of the kynurine pathway in aging, and identifies that downregulation of a fourth member of this pathway, the enzyme HAAO, extends lifespan in *C. elegans* by increasing the levels of 3HAA, the HAAO substrate. Additional experiments in *C. elegans* allow the authors to link 3HAA elevation and tryptophan metabolism with reduction of oxidative stress and activation of the cellular Nrf2/SKN-1 oxidative stress response, which constitute a well-characterized pathway in this model with strong implications in the aging process.

Furthermore, the authors complement their experiments in preclinical model, and show that feeding aged mice (27mo) with 3HAA also extend rodent lifespan, which is accompanied by a less inflammatory immune profile. Overall, the authors propose that HAAO and 3HAA can be considered as novel therapeutics targets to combat aging and/or age associated diseases.

The article is written in excellent English and aligns with the growing evidence that characterizes tryptophan metabolism and associated metabolites, including NAD⁺, as important mediators of aging and potential targets with lifespan increasing properties. However, there are several concerns with regards to the accuracy and interpretation of specific data sets. Additional experiments are also required to certify the specificity of 3HAA-mediated effects. Other recommendations are also included to improve the quality of the manuscript.

With regards to the mouse data sets, the findings are to some extent interesting. However, the sample size constitutes a strong limitation, with the physiological benefits reported being very mild. In addition, the authors do not validate the levels of 3HAA, the

activation/repression of the Tryptophan metabolism and/or the cellular antioxidant Nrf2-reponse. In the context of the manuscript, this information seems required to comprehend the molecular insights that 3HAA modulates in the mouse aging.

From all of the above, please find the following specific comments:

Lines 77-78 (1st comment): The authors claim that knockdown of the gene encoding HAAO, *haao-1*, robustly extended *C. elegans* lifespan by increasing physiological levels of (3HAA), a mechanism DISTINCT from inhibition of *tdo-2*, *kynu-1*, or *acsd-1*. This is a bit surprising because extended figure 1B shows a very strong and quite similar regulation of certain metabolites involved in central pathways influencing aging such as NAD and/or TRP. Similarly, KA and AA are strongly induced and reduced in the three RNAi models. 3HAA is also enhanced in *tdo-2*(RNAi) animals, although to a lesser extent than in *haao-1* (RNAi). Therefore, shared mechanisms of action mediated by these commonly regulated metabolites may exist and may be responsible for lifespan extension and enhancement of health beyond the effects of 3HAA. Please incorporate this rationale in the context of the manuscript and include in the discussion their relevance and potential impact in the context of aging of these other metabolites. In order to corroborate (or invalidate) 3HAA-mediated specificity, it would be helpful to identify associations between the levels of these metabolites and lifespan extension using data already collected by the authors.

Lines 77-78 (2nd comment): the authors claim that, among the distinctive mechanisms from inhibition of *haao-1* includes increased resistance to oxidative stress and *skn-1* activation (shown in Figure 3). To confirm this specificity, please determine that *kynu-1*(RNAi) and *tdo-2*(RNAi) shows reduced stress resistance and lower activation of cellular oxidative stress response when compared with *haao-1*(RNAi) and/or the null mutation *haao-1(tm4627)*.

Lines 93-98, related to Figure 1C and D: downregulation of TM pathway by *kynu-1*(RNAi), *tdo-2*(RNAi), and *haao-1*(RNAi) reduces worm area and motivated/unmotivated speed early in life but not late in life. Please discuss the relevance of activation/repression of tryptophan metabolism at different age-onsets. It would be helpful to show how 3HAA supplementation late in life in *C. elegans* also extends lifespan. The same applies for TOD2 inhibitors and/or

potential HAOO inhibitors if they are available. Please show the effects when the repression starts late in life. Of relevance, metabolite quantification was performed on day 4 of adulthood (i.e. when worms were smaller and with reduced speed). Therefore, please corroborate that 3HAA levels remain higher late in life when the health-related benefits are found.

Line 111-120, related to Figure 1G and Extended Figure 3: The authors examined the interaction between kynurenine metabolism and genetic pathways with established roles in aging (*daf16*, *eat2*, *hif1*, *rsk-1*, *sir2-1*). It is quite surprising that none of these mutants show extended lifespan (Extended Figure 3), which directly opposes with numerous publications in the field of aging. Therefore, inferring the impact of *kynu-1*(RNAi), *haao-1*(RNAi), and *tdo-2*(RNAi) on lifespan over these strains can't simply be done or wouldn't be acceptable. Do the authors have an explanation?

Line 136-137, related to Figure 2E: the null mutation *haao-1(tm4627)* mimics the lifespan extension mediated by *haao-1*(RNAi). At this point, I kindly suggest to the authors to provide the data to whether the null mutation *haao-1(tm4627)* behaves the same for the other parameters analyzed (i.e. lifespan extension at different viable temperatures, worm area, speed, thrashing, total progeny, egg production, tryptophan-related metabolite levels (including 3HAA), etc etc...). Although apparently trivial, this is of relevance for the manuscript to corroborate their conclusions.

Line 93, related with Figure 1A and Extended Figure 3 and 4: "Like *kynu-1*(RNAi) but unlike *tdo-2*(RNAi), this benefit was consistent across the viable temperature range for *C. elegans*". This assumption seems clear in Figure 1A. However, it seems that lifespan extension is much stronger at 15C than at 25C in extended figures 3 and 4. Would it be possible to quantify the impact of temperature on *haao-1*(RNAi)-mediated lifespan extension from these extended panels? How the authors reconcile these two different observations?

Line 197-198, related to Figure 3: "Indeed, lifespan extension from *haao-1*(RNAi) was absent in *C. elegans* with the hypomorphic *skn-1(zj15)* mutation (Fig. 3G)". Very interesting data.

Following the experimental design of figure 3, 3HAA supplementation shouldn't extend lifespan in *skn-1* (zj15) animals. Likewise, lifespan extension in the null mutation *haao-1*(tm4627) should be absent in *C. elegans* with the hypomorphic *skn-1*(zj15) mutation. Please, validate these sets of experiments.

Line 209-217, related to Figure 4 and Extended Figure 8: The authors show mild physiological benefits on aged mice after 3HAA supplementation. Since Rotarod, Grip Strength, immune populations, blood glucose, TG, etc.. were recorded both at baseline and after treatment, I kindly suggest calculating the effects of 3HAA as the %change from baseline at individual level. This analysis will determine if 3HAA-mediate a beneficial effect on the several metrics of health. Please show these representations for the panels included in figure 4 and extended figure 8.

Line 211-212: Please normalize the latency to fall per gr of body weight. Do the differences remain?

Line 217, related to extended data 8B: Please analyze glucose-clearance trajectories as %change from time 0.

Line 245, related to statistical significance: the authors claim that dietary 3HAA promotes lifespan extension based on a Cox-regression method to violate the proportional hazard assumption. However, for the specific experimental set/design, in which mice are supplemented with 3HAA late in life (27mo old), a proportional hazard ratio (i.e. same risk of dying) should be in my opinion applied and/or assumed. In lines 466-467, it is not clear to me whether the 'failure' of the Cox proportional hazard regression model means that the statistical test could not be applied or, on the other hand, if the test was run and statistical significance was not found. In addition, the estimation of the survival function/time cannot be achieved with this model. Therefore, I am kindly asking to provide the survival time and statistical evaluation using the Kaplan-Meier model

Line 245: the authors claim that dietary 3HAA is sufficient to promote healthy longevity in

the mouse model. How the authors explain this effect? How is 3HAA metabolized? Are the levels of 3HAA increased? Are the levels of other tryptophan-metabolites modulated? Does 3HAA supplementation reduce oxidative stress and activates cellular Nrf2 response? All these questions remain unresolved. In my opinion, the activation of the main 3HAA-mediated mechanisms should be proved, especially when the sample size for these set of experiments is very small. In the end, the authors should demonstrate how 3HAA is acting in rodents.

Additional recommendations:

Line 94-95, relative to Figure 1C: the authors indicate that *haao-1*(RNAi) animals were slow to reach full body size (Fig 1G), but resistant to the age-dependent reduction in body size observed in animals subjected to empty vector (EV) RNAi. I disagree with this comment since *haao-1*(RNAi) never reaches full body size (it is lower than controls at 7, 14 and 21 days at 15C; and also lower than controls at every time-point analyzed at 25C]. It would be helpful to understand if *haao-1*(RNAi) and/or 3HAA supplementation impacts well-characterized metabolic pathways that governs cell growth (mTOR pathway, AMPK,..).

Line 99-100, related to Fig. 1E: “Total brood size was not impacted by *haao-1*(RNAi) (Fig. 1E); however, egg production was slightly delayed relative to EV(RNAi), similar to *kynu-1*(RNAi) and less pronounced than *tdo-2*(RNAi) (Extended Data Fig. 1A)”. In my opinion, caution should be used here if these enzymes/metabolites impact fertility. Please extend the discussion of this topic accordingly, which confronts with the potential role of HAOO and 3HAA as novel therapeutics targets to combat aging. Including these negative-side effects on the abstract is recommended.

Line 206-209, related to Fig 4A: In my opinion, Fig 4A-top panel is, in my opinion, not accurate, since it seems that mice were randomized into the experimental groups at birth and any of them died birth to at least 27 months of age. Rather, mice at the age of 27 months (survivors) were divided into the experimental groups. I would recommend to remove Fig 4A-bottom panel and incorporate age (in weeks) together with Time on diet (in weeks) on Fig4A-bottom panel.

Line 137-140: “If our model is accurate and 3HAA mediates benefits of decreasing HAAO activity, but not KYNU or TDO2, then 3HAA should further increase the already long lifespans of animals with reduced kynu-1 or tdo-2, but not animals with reduced haao-1. Our observations confirmed this prediction (Fig. 3F, Extended Data Fig. 6)”: I think this data set do not correspond to Figure 3F and Extended Fig 6 do not show that data.

Line 206: please include the number of mice in this sentence (i.e. n = 6-8 mice per group)

Lines 257-259: Since the authors highlight that haao-1(RNAi), kynu-1(RNAi) and tdo-2(RNAi) reduces NAD levels, while elevating NAD is shown to be beneficial for healthy aging, could the authors show if acetylation (as a function of reduced activity of the NAD-consuming enzyme Sir) is increased?

Reviewer #3 (Remarks to the Author):

The paper by Dang et al. analyzed the effects of knocking down the KP enzyme haao-1, and supplementing with 3HAA, the substrate of haao-1, in *C. elegans*. Their study uses a mix of phenotypic assays (lifespan, motility, fecundity) and biochemical assays (metabolite MS, H₂O₂, GFP reporters) to conclude that knockdown of haao-1 increases lifespan via the accumulation of its substrate 3HAA. They show that treatment with 3HAA has an anti-oxidant effect, and activates the oxidative stress response in *C. elegans*. They finally validate their findings in a small cohort of mice to show that 3HAA has a similar effect on lifespan, as well as improvement in health and behavioral functions. The study is of interest to the anti-ageing field as it suggests that 3HAA may be a promising therapeutic target to target the effects of ageing. The manuscript is well-written and the data are clear and convincing. With a few small changes we can recommend this manuscript for publication.

In Figure 2 and Extended Figures 5,6 the authors show evidence that 3HAA supplementation mediates haao-1 lifespan extension. In extended Data Figure 1B, the haao-1 RNAi also results in an increase in various other metabolites including tryptophan, kynurenic acid and cinnabaric acid. A useful additional experiment would be to also supplement with these additional metabolites and perform lifespan assays to further elucidate potential

mechanisms of *haao-1* knockdown. Additionally, to further show that 3HAA supplementation can replicate the effect of *haao-1*, it would be fruitful to also check motility assays in response to 3HAA in a separate phenotypic assay (as in Figure 1).

In Figure 3 and Extended Figure 7, the authors show that 3HAA boosts oxidative stress resistance. Figure 3B would be improved with some element of quantification, such as a spectrophotometric assay. It would also be useful to check motility of *haao-1* RNAi in a *skn-1* mutant background (Figure 3G) to see if the effect is the same in an additional phenotypic assay.

General notes: many replicate number and sample sizes not clearly indicated for many of the experiments in the main text, figure legends, or materials and methods. There are a few typos: in Line 144, should be 2F instead of 3F, in Figure 4D “should be Inflammatory” instead of “inflammmitory”. Some figures duplicated in the main figures and in the extended figures, e.g. Figure 2D and Extended Figure 5C.

RESPONSE TO REVIEWER COMMENTS

We would like to start by thanking each of the reviewers for providing constructive critiques of this work. There are some complexities and caveats to 3HAA as a therapeutic target and the reviewers brought up several that are important considerations which we had not explicitly addressed in the original manuscript. We address each comment below.

As a broader note that is relevant to our response across many of these points, we have continued to study 3HAA in several contexts—cancer, immunity, stress response—since submitting this manuscript, and some of the experiments requested by the reviewers were underway during the review. With specific regard to the mechanism of lifespan extension by 3HAA. Our current model is that the mechanism of action is multifaceted and cannot be captured by a single pathway. This is reflected in the partial prevention of the lifespan extension from *haao-1* knockdown by combination with mutations in five common aging pathways (Figure 1G). Knockdown of *skn-1* was the most robust, and thus received the most focus in the manuscript. This continues to be the main message of the manuscript, but we have modified our discussion of the relevant findings to more clearly communicate that the mechanism is not as simple as just activation of SKN-1. We do not see this as unique to the interventions examined in this work. In general, we find that publications that attempt to narrowly focus on a single mediating mechanism as the “whole story” for a given intervention in the context of aging tend to miss the broader picture. We frame the interaction between *haao-1* and *skn-1* as a starting point, which will be expanded up on in future papers.

To that point, we have done a lot of work to add detail to our understanding of the mechanisms linking *haao-1* and 3HAA to aging. There simply is not sufficient space in a single manuscript with strict length requirements to fully describe these mechanisms, and we have two additional full manuscripts nearly complete that describe aspects of these mechanisms in additional detail, each of which is dependent on publishing the work outlined here. In our response to the comments, we highlight places where we have made modifications to this work to address the comments, and places where the detail of the full mechanism is too complex for a simple adjustment to this work and will be address more fulling in upcoming manuscripts. We hope this is sufficient to satisfy the critiques.

Reviewer #1 (Remarks to the Author):

The work “3-hydroxyanthranilic acid– a new metabolite for healthy lifespan extension” authored by Dang H. et al present data supporting a role for 3-hydroxyanthranilic acid (3HAA) in lifespan extension in *C. elegans*. Authors suggest the anti-ageing effect of 3-HAA is achieved due to the antioxidant properties of 3-HAA -both directly by scavenging H₂O₂ and indirectly by activating SKN-1 (NRF2 homolog). An experiment in mice is provided aiming at generalizing the finding reported in *C. elegans*. Overall the manuscript is well written and further explores a metabolic pathway that has been related to lifespan extension in previous reports. The authors present convincing data

supporting the effect of 3-HAA on lifespan extension. On the other hand, their claim that 3HAA is “healthy” might contradict previous reports. I have also found not very convincing the mechanism(s) by which 3-HAA is driving life extension. The following issues should be addressed to clarify the mechanism and the reactivity of 3HAA:

To the note about whether this can be considered a “healthy” intervention. This claim is based on the observation that worms treated with 3HAA exhibit improved health across several metrics associated with movement later in life compared to experimentally matched controls. These are the most common measurements of health in this model system. Our claim is simply that the net impact of elevating 3HAA in the context tested is longer lifespan and improved health later in life. Our continuing investigation of 3HAA suggests that the underlying mechanisms are complex and non-singular. To the reviewer’s point, 3HAA is likely having intermediate effects that are both beneficial and detrimental and like many compounds. The improved health results from the benefits outweighing the negatives. While this does detract from the therapeutic potential in some sense (aka 3HAA will not be a perfect solution), the same is likely to be true of any intervention that is capable of intervening in the aging process, and indeed most pharmaceutical interventions of any type. At least, we are not aware of any single intervention that is free of some negative intermediate consequence (e.g., dietary restriction, rapamycin, senolytics, NAD precursors, antibiotics, aspirin all have downsides to consider). Given that the net impact of the intervention on downstream consequences of functional health, we don’t see this as a reason to specifically penalize 3HAA relative to any other promising intervention in the aging space. We address specific instances below.

We do appreciate the reviewer’s point that overuse of terms like “health” and “healthspan”, which are widely and non-specifically used in the literature. We edited several points in the manuscript to limit overuse and be more precise in our claims, and changed the title to remove “healthy”. See the next point for more specifics on 3HAA as a mutagen/carcinogen.

Major issues.

1- 3-HAA has been previously reported to significantly impair tumor growth (doi: 10.1038/s41420-021-00561-6), and it was originally described as an endogenous mutagen (<https://doi.org/10.1038/177837a0>). The translational application of 3HAA might be limited considering the mutagenic and cytotoxic effect of 3HAA reported on mammalian cells. Therefore, how authors reconcile their proposal for of 3HAA as therapeutic intervention? Is 3HAA a mutagen in your setting? They might perform toxicity and/or mutagenicity assays in cells to address this concern.

The historical data suggesting mutagenic properties of 3HAA is an important point and worth consideration. We added a section to the revised Discussion outlining the mixed evidence for 3HAA as a mutagen and carcinogen. As the reviewer suggested, we include new cytotoxicity assays in both HEK293 and 3T3 cells (added to Extended Data Fig. 5). We also include kynurenine pathway metabolite data from both plasma and urine in our mouse studies and discuss the 3HAA dose experienced by cells in our

mouse studies in the context of both the cytotoxicity assays and the past studies. In brief, our reading of the available data suggests to us that the effect of 3HAA on tumorigenesis depends on dose. The early studies showing that 3HAA can induce tumor formation used a solid pellet of 3HAA in the bladder, and likely produced very high local concentrations that are well above the therapeutic doses employed by more recent studies that show benefit, including this one.

2- In Fig. 1F the results show similar overall life change in *kynu-1* and *haao-1*. If accumulation of 3HAA is driving the phenotypes, then 3HAA accumulation in *HAO-1* siRNA might be causing product inhibition of *kynu-1*, thus leading to phenotype. The Fig. 2F aims at resolving the epistatic interaction among *kynu-1*, *haao-1* in presence of 3-HAA. The control should be shown in each plot, helping the comparison. Authors should also perform/show ageing experiments to compare single and double knock-downs in absence of 3-HAA (like Fig. 1B but with double knockdown/out).

The interaction between 3HAA, *haao-1*, and *kynu-1* is an important aspect of this work and is indeed the purpose of Fig. 2F (now Fig. 2H). We did not include all of the controls in the primary figure to avoid increasing the complexity of the images (for readability). We did include all epistatic interactions between *tdo-2*, *haao-1*, and *kynu-1* in the original manuscript, and the full figure panels showing the impact of 3HAA in each single knockout/knockdown strains and in the *haao-1 kynu-1* double mutant (with all controls) in Extended Data Fig. 6 in the original submission (now Extended Data Fig. 8). The statistics for all pairwise comparisons are also provided in Extended Data Table S9. These summary statistics include each replicate experiment and the pooled data across replicates.

If we are misunderstanding the request and this is not what the reviewer is asking, we are happy to include more detail or additional comparisons.

The reviewer's suggestion that 3HAA may be inhibiting KYNU is a good one. This is supported, in part, by additional data that demonstrates a consistent downregulation of anthranilic acid (AA) in worms in response to *haao-1* inhibition, and a dose-dependent reduction in lifespan when AA is supplemented. This data is now included in Extended Data Fig. 6 (along with data on supplementing several other metabolites to address other comments made below) and referenced in Results. We also added a couple of sentences to discuss this point in the second paragraph of the Discussion.

3- From the data presented in Fig. 3, it looks like 3HAA-anti-aging effect is more pronounced in absence of *kynu-1* than in absence of *haao-1*. Hence, 3HAA might somehow enhance the effect of *kynu-1* deletion, or as proposed, work through a completely different mechanism. Expanding the discussion of the role of *kynu-1* would help to clarify the possible mechanisms of 3-HAA anti-aging. In addition, a clear additive effect in anti-ageing would be expected by growing worms with TRP and 3HAA -further supporting independent effects.

These are good suggestions and areas of our ongoing interest. One issue here is that we do not know the mechanism through which *kynu-1* extends lifespan. In our metabolite data we find that KYN, but not 3HK, is consistently upregulated by knockdown of *haao-1* in *C. elegans*, though to a much lesser extent than knockdown of *kynu-1*, and the upregulation appears to decrease with age (see metabolite data in Extended Data Fig. 6). We tried supplementing worms with KYN and found a slight reduction in lifespan. As we note above, AA is also down in response to both *kynu-1* and *haao-1* knockdown animals, and AA also shortens lifespan. Thus, restriction of AA production by inhibition of KYNU is one possible mechanism of action supported by our current data, and one that is not mutually exclusive with NRF2/SKN-1 activation. We added discussion to this effect.

For the suggested TRP experiments, we agree that this is an interesting direction. Our attempts to supplement TRP have been highly variable and produced inconsistent lifespan effects across different experimental repeats. There is one published study (PMID: 25643626) that reports lifespan extension following 1 mM or 5 mM tryptophan supplementation in *C. elegans*; however, this paper conducted experiments in liquid culture. Worms are known to respond differently to interventions that extend lifespan in liquid vs. solid culture. We find that tryptophan supplementation at 1 mM or above induces developmental arrest in a temperature-dependent manner (with a greater effect at 25°C). We suspect that tryptophan produces consequences beyond its role as a substrate for kynurenine metabolism that may confound these experiments. We are still troubleshooting this problem and don't have publishable data at this stage but hope to address this issue in the near future.

4- Authors claim the 3HAA effect might be independent of KYN. 1) they claim 3-HAA is redox active and induces the degradation of H₂O₂. A more in vivo demonstration of this effect would be to use the HyPer worms (see <https://doi.org/10.1038/s41467-022-28242-7>).

5- Authors show that *haao-1* inactivation upregulates SKN-1 (NRF2) in worms. SKN-1 is usually activated by compounds that cause oxidative stress (see <https://doi.org/10.1038/s41467-022-28242-7>), thus it would be expected that the accumulation of 3-HAA in *haao-1* knockdown could lead to repression of SKN-1 instead of activation. Does 3HAA show additive effect with other compounds able to modulate SKN-1?

6- Related to the previous issue, 7- SKN-1- mammalian ortholog (NRF2) is under the control of KEAP1. In worms, KEAP1-ortholog has not been described, and instead the oxidative stress response is triggered through a signaling cascade that results in the phosphorylation and activation of the MAPKK and P38/MAPK homologs. Are those signalling kinases also phosphorylated in response to 3HAA or in *haao-1* mutant? Could authors speculate how SKN-1 would be activated by 3HAA in worms?

7- A simple test supporting a general role of 3HAA in regulating SKN-1 would be to measure NRF2-target genes in a mammalian cell in presence of 3HAA.

As points 3 – 7 all relate to SKN-1 and 3HAA, we are combining our responses. Yes, these are all good points and something that we were wondering about as well. In short, we have done a lot more work on how SKN-1 is activated in worms with elevated 3HAA. Our current model is that 3HAA activates SKN-1 in part by elevating ROS and in part through a non-canonical mechanism. The complete story is in preparation as a separate manuscript that we hope to submit shortly after this work is published. This work is more focused on oxidative stress resistance and the mechanisms linking 3HAA to SKN-1, and contains too many experiments to include here, given the space constraints.

However, we do agree that additional data is warranted in this work to clear up the mismatch between the decrease in hydrogen peroxide by 3HAA and activation of SKN-1. As the reviewer suggests, we did look at the worms with HyPer to determine what is happening *in vivo*. It seems that endogenous hydrogen peroxide is, in fact, increased internally, despite the *in vitro* impact of 3HAA on hydrogen peroxide and the reduced excretion with age. Two recent papers independently report that 3HAA activates NRF2 in cell culture (which we reference in the Results; PMID: 36627132, PMID: 35245456), and we further include a Western blot confirming activation of SKN-1 in mammalian cells in Extended Data Figure 8C. We briefly discuss possible mechanism of SKN-1 activation by 3HAA in light of this new data in the discussion.

8-The epistatic analysis of SKN-1 *haao-1* (Fig. 3g) does not contribute to clarify the relationship between *skn-1* and *haao-1* because it only said that the effect of *skn-1* inactivation is dominant. A better experiment would be to show the lifespan but adding 3-HAA. This experiment would indicate if 3-haa effect is independent of *skn-1* or necessitates of *skn-1* (like Extended Fig. 6 with SKN-1 *haao-1* strain)

We agree and conducted this experiment, along with additional replicates combining *skn-1* mutants with *haao-1(RNAi)* (the original submission included only two replicates for the latter). With these additional replicates, and the 3HAA data, it appears that the prevention of lifespan extension in the *skn-1* mutant is likely incomplete. In the case of both 3HAA and *haao-1(RNAi)*, it appears that lifespan extension is largely, but not completely, abrogated in the *skn-1* background. One possible explanation is that the *skn-1* mutant is a hypomorph, not a complete loss of function, so residual SKN-1 activity may explain the small lifespan extension. However, this mutant does prevent lifespan extension from other interventions, so we see this as unlikely. Given the impact of *haao-1(RNAi)* on lifespan other aging mutants (Figure 1G), we believe that the complete mechanistic picture for 3HAA lifespan extension is more complex than just a single pathway. This is a topic of ongoing work. We have modified and expanded our SKN-1 discussion to reflect this new data.

Minor issues

1 -Fig. 1B, the legend is grey and the line black.

Corrected. Thank you.

2- Line 140, should be Fig. 2F

Thank you for catching the error; this was a mis-reference in the original manuscript. In the current version these data are Figure 2H. The text should now be correct. We also noted a reference to Fig. 2E a few lines earlier should have been Fig. 2D, which is now corrected.

3- Extended Data Fig. 5. The phrase “3HAA is upregulated” is not appropriate. May be use “3HAA accumulation” instead.

Yes, “3HAA accumulates” is more accurate phrasing. Thank you and corrected.

Reviewer #2 (Remarks to the Author):

Based on previous published work from the same group, this manuscript supports the relevance of the kynurenine pathway in aging, and identifies that downregulation of a fourth member of this pathway, the enzyme HAAO, extends lifespan in *C. elegans* by increasing the levels of 3HAA, the HAAO substrate. Additional experiments in *C. elegans* allow the authors to link 3HAA elevation and tryptophan metabolism with reduction of oxidative stress and activation of the cellular Nrf2/SKN-1 oxidative stress response, which constitute a well-characterized pathway in this model with strong implications in the aging process. Furthermore, the authors complement their experiments in preclinical model, and show that feeding aged mice (27mo) with 3HAA also extend rodent lifespan, which is accompanied by a less inflammatory immune profile. Overall, the authors propose that HAAO and 3HAA can be considered as novel therapeutics targets to combat aging and/or age associated diseases.

The article is written in excellent English and aligns with the growing evidence that characterizes tryptophan metabolism and associated metabolites, including NAD⁺, as important mediators of aging and potential targets with lifespan increasing properties. However, there are several concerns with regards to the accuracy and interpretation of specific data sets. Additional experiments are also required to certify the specificity of 3HAA-mediated effects. Other recommendations are also included to improve the quality of the manuscript.

With regards to the mouse data sets, the findings are to some extent interesting. However, the sample size constitutes a strong limitation, with the physiological benefits reported being very mild. In addition, the authors do not validate the levels of 3HAA, the activation/repression of the Tryptophan metabolism and/or the cellular antioxidant Nrf2-reponse. In the context of the manuscript, this information seems required to comprehend the molecular insights that 3HAA modulates in the mouse aging.

From all of the above, please find the following specific comments:

Lines 77-78 (1st comment): The authors claim that knockdown of the gene encoding HAAO, *haao-1*, robustly extended *C. elegans* lifespan by increasing physiological levels of (3HAA), a mechanism DISTINCT from inhibition of *tdo-2*, *kynu-1*, or *acsd-1*. This is a bit surprising because extended figure 1B shows a very strong and quite similar regulation of certain metabolites involved in central pathways influencing aging such as NAD and/or TRP. Similarly, KA and AA are strongly induced and reduced in the three RNAi models. 3HAA is also enhanced in *tdo-2*(RNAi) animals, although to a lesser extent than in *haao-1* (RNAi). Therefore, shared mechanisms of action mediated by these commonly regulated metabolites may exist and may be responsible for lifespan extension and enhancement of health beyond the effects of 3HAA. Please incorporate this rationale in the context of the manuscript and include in the discussion their relevance and potential impact in the context of aging of these other metabolites. In order to corroborate (or invalidate) 3HAA-mediated specificity, it would be helpful to identify associations between the levels of these metabolites and lifespan extension using data already collected by the authors.

This is a good point. We agree that the metabolite data suggests that a shared mechanism between knockdown of *tdo-2*, *kynu-1*, and *haao-1* may be evident beyond 3HAA. Our original metabolite data was conducted in young worms (day 4 of adulthood) at 15°C. We note further that TRP and KYN are also upregulated by *haao-1*(RNAi), though to a lesser extent than *kynu-1*(RNAi). We first looked at KYN, KA, and AA levels in a new set of young (day 4 adulthood) and old (day 12 adulthood) worms at 20°C. In this case, the elevated TRP and KA levels in response to *haao-1* and *kynu-1* knockdown from the first experiment did not replicate (indicating that this is not a likely mediator across environmental contexts), but the elevated KYN and the reduced AA levels did replicate. As a follow up, we exposed worms to KYN and AA to determine the potential impact on lifespan. KYN supplementation slightly reduced lifespan, inconsistent with elevated KYN mediating lifespan extension. However, AA also reduce lifespan in a dose-dependent manner. This is consistent with a model in which reduced AA is a common mechanism of lifespan extension in response to knockdown of *tdo-2*, *kynu-1* and *haao-1*, as the reviewer suggests. Further work is needed to fully validate this model, which we intend to pursue given this new data. We note that because 3HAA further extends the lifespan when *kynu-1* or *tdo-2* is knocked down, 3HAA elevation and AA reduction are likely parallel mechanisms. The new metabolite measurements and lifespan data are included in Extended Data Fig. 6 in the revised manuscript. We also added discussion of the potential for cross-regulation between different parts of the kynurenine pathway per the reviewer's suggestion in the Discussion.

Lines 77-78 (2nd comment): the authors claim that, among the distinctive mechanisms from inhibition of *haao-1* includes increased resistance to oxidative stress and *skn-1* activation (shown in Figure 3). To confirm this specificity, please determine that *kynu-1*(RNAi) and *tdo-2*(RNAi) shows reduced stress resistance and lower activation of cellular oxidative stress response when compared with *haao-1*(RNAi) and/or the null mutation *haao-1*(tm4627).

We may have given the wrong impression on this point. Our model is that the upregulation of 3HAA is specific to *haao-1* knockdown, and that 3HAA influences lifespan (at least in part) by activating NRF2/SKN-1. This model is agnostic to whether either *kynu-1* or *tdo-2* knockdown work through activation of SKN-1; however, if SKN-1 does mediate the effect of *kynu-1* and *tdo-2* knockdown, it must do so through a mechanism independent of 3HAA, since 3HAA is not elevated. Indeed, 3HK, which does accumulate in *kynu-1* mutants, shares many of the context-dependent (e.g., pH and metal) redox properties with 3HAA, and may similarly activate NRF2/SKN-1. Indeed, two recent papers suggest that KYN and 3HK can also activate NRF2 in mammalian cell culture (PMID: 36627132, PMID: 35245456). For these reasons, the state of oxidative stress resistance and SKN-1 activation state in animals with reduced *tdo-2* and *kynu-1* do not have direct bearing on our model, and we consider these experiments out of scope for this work. Beyond this manuscript, we are very interested in pursuing the mechanisms mediating *kynu-1* once we obtain additional funding (which is currently a primary limiting factor).

Lines 93-98, related to Figure 1C and D: downregulation of TM pathway by *kynu-1*(RNAi), *tdo-2*(RNAi), and *haao-1*(RNAi) reduces worm area and motivated/unmotivated speed early in life but not late in life. Please discuss the relevance of activation/repression of tryptophan metabolism at different age-onsets. It would be helpful to show how 3HAA supplementation late in life in *C. elegans* also extends lifespan. The same applies for TOD2 inhibitors and/or potential HAAO inhibitors if they are available. Please show the effects when the repression starts late in life. Of relevance, metabolite quantification was performed on day 4 of adulthood (i.e. when worms were smaller and with reduced speed). Therefore, please corroborate that 3HAA levels remain higher late in life when the health-related benefits are found.

Removing potential negative early-life consequences of interventions that extend lifespan is an important consideration, particularly during development. These experiments were underway at the time of submission. We now include experiments showing that 3HAA supplementation and *haao-1*(RNAi) initiated during adulthood extend lifespan to a similar degree as when they are started at egg (Extended Data Fig. 5C-E). We further show that the 3HAA analog 4CI-3HAA, a HAAO inhibitor, increases lifespan in a dose-dependent manner when initiated during adulthood. The degree of extension was not as great as experiment-matched worms treated with *haao-1*(RNAi), but the maximum observed lifespan extension was at the lowest tested dose, so we likely have not found the optimal 4CI-3HAA dose (Extended Data Fig. 6N). We further show that 3HAA continues to increase in concentration late in life in worms lacking *haao-1* (Extended Data Fig. 6A).

Line 111-120, related to Figure 1G and Extended Figure 3: The authors examined the interaction between kynurenine metabolism and genetic pathways with established roles in aging (*daf16*, *eat2*, *hif1*, *rsks-1*, *sir2-1*). It is quite surprising that none of these mutants show extended lifespan (Extended Figure 3), which directly opposes with numerous publications in the field of aging. Therefore, inferring the impact of *kynu-1*(RNAi), *haao-1*(RNAi), and *tdo-2*(RNAi) on lifespan over these strains can't simply be

done or wouldn't be acceptable. Do the authors have an explanation?

There is a complication with comparing the Extended Data Figure 3 to published work. Historically, *C. elegans* work has been conducted at 20°C (the majority of studies) or 25°C (fewer studies, but still relatively common). A very small minority examine animals at 15°C, likely because the animals are substantially longer-lived and experiments take more time at this temperature. We have a 2017 Aging Cell paper that explores the role of temperature in *C. elegans* lifespan in more detail (PMID: 28940623), with a specific emphasis on the interaction between temperature and common genetic determinants of longevity. In brief, many genetic interventions that extend lifespan at 20°C have a modified effect at 15°C. Our data generally agree with published work for the five aging mutants examined (*daf-16*, *eat-2*, *hif-1*, *rsk-1*, *sir-2.1*), considering environmental context as a relevant variable, with the notable exception is *sir-2.1*. Here we provide a short synopsis of each gene examined in this manuscript (please reference Extended Data Figure 3 for 15°C survival curves, Extended Data Figure 4 for 25°C survival curves and Supplemental Table S6 for statistics):

- *daf-16* encodes the FOXO family transcription factor that is activated in response to reduced insulin signaling (among other upstream processes). *daf-16* knockdown is well-known to shorten lifespan at 20°C and 25°C, and according to our above referenced paper (PMID: 28940623), it does not change lifespan at 15°C. This is in agreement with our data (Extended Data Figures 3 and 4, top row).
- *eat-2* mutants are an established model of dietary restriction resulting from decreased pharyngeal pumping, which results in reduced food intake. This model is widely reported to be long-lived across many papers (though to our knowledge, this has not been tested at 15°C previously). In this work, the *eat-2* mutants were long-lived at both 15°C (Extended Data Figure 3, second row) and 25°C (Extended Data Figure 4, second row), the latter of which is in agreement with published work.
- *hif-1* encodes the hypoxia inducible factor 1 (HIF1), a transcription factor activated in response to hypoxic signaling. *hif-1* mutants have a known interaction with temperature in *C. elegans* (see PMID: 21241450, PMID: 28940623). In fact, a difference in environmental temperature turned out to be the resolution to apparently contradictory data published by the Kaerberlein and Kapahi labs following the early work on hypoxic signaling in *C. elegans* aging (PMID: 19633411). According to the above papers, *hif-1* knockdown extends lifespan at 25°C, but not at 15°C, which is in agreement with our data in this work.
- *rsk-1* is the *C. elegans* ortholog of ribosomal protein S6 kinase and has been previously reported to extend lifespan at 25°C (PMID: 24332851) and shorten lifespan at 15°C (PMID: 28940623). Results are mixed at 20°C (PMID: 17266680, PMID: 28940623, PMID: 33274335, PMID: 17266680). In this work, *rsk-1* extended lifespan at 25°C and either had no effect, slightly extended, or slightly shortened lifespan depending on the experiment at 15°C. These results

are on par with published data, taking the experimental variability in this strain into account as a caveat (which does limit interpretation for this strain at this temperature).

- *sir-2.1* is the *C. elegans* ortholog of yeast *sir2* and human SIRT1 and knockout or knockdown is widely reported to have no effect (PMID: 19370397, PMID: 23509272, PMID: 21938067, PMID: 35096951) or slightly shorten lifespan (PMID: 16777605, PMID: 21909281) at 20°C. We found one case where lifespan for *sir-2.1* knockout worms was measured in the long-lived hypomorphic *daf-2(e1370)* background, and in this case *sir-2.1* deletion extended lifespan relative to worms with *daf-2(e1370)* allele alone (which is possibly confounded by interaction with the *daf-2* allele; PMID: 16777605), and another case where *sir-2.1* knockout animals appeared longer-lived (mean lifespan 22.7 days) vs. wild type (mean lifespan 18.9 days) at 25°C, though a statistical comparison was not conducted between these two strains (PMID: 28239780). A third study reports that *sir-2.1* mutants are short-lived at 25°C (PMID: 19164523). In our previous work we found that *sir-2.1* knockout worms are long-lived at 15°C (PMID: 24764514). In this work, we found *sir-2.1* knockout worms to be not different from wild type at 15°C, which is different from our earlier finding, and slightly long-lived at 25°C, in agreement with two published papers at this temperature, and in disagreement with a third. This is the one case where our data are not fully aligned with the published literature, but also a case where the available data on the interaction between *sir-2.1* and temperature is sparse and highly variable. More work will be needed to understand this point, and we agree that a strong conclusion from our interaction data is not warranted.

Given this information and the agreement of our data with the literature in almost all cases, we feel that our interpretation of the interaction data was appropriate. If there are specific cases that we missed where our data is not in agreement with established trends in the literature, we will be happy to examine these points and adjust our interpretation and discussion appropriately. We ask that specific references be provided if this is the case.

Line 136-137, related to Figure 2E: the null mutation *haao-1(tm4627)* mimics the lifespan extension mediated by *haao-1(RNAi)*. At this point, I kindly suggest to the authors to provide the data to whether the null mutation *haao-1(tm4627)* behaves the same for the other parameters analyzed (i.e. lifespan extension at different viable temperatures, worm area, speed, thrashing, total progeny, egg production, tryptophan-related metabolite levels (including 3HAA), etc etc...). Although apparently trivial, this is of relevance for the manuscript to corroborate their conclusions.

This is a useful addition. We also note that the majority of the 3HAA data is at 20°C, while the data on worm health parameter were only reported at 15°C and 25°C in the original submission. We now include brood size (Figure 2F, Extended Data Figure 5B) and thrashing (Figure 2G) for a set of experiments that included direct comparison between control, *haao-1(RNAi)*, *haao-1(tm4627)*, and 1mM 3HAA supplementation at 20°C. We note that the original data was collected while Dr. Sutphin was at The

Jackson Laboratory (using the WormLab system in the laboratory of a collaborator at Mount Desert Island Biological Laboratory), and he has since moved to the University of Arizona and does not have access to that system. Thus, we did not have a simple means to collect the more sophisticated movement data. We also include metabolite data (Extended Data Figure 1C) for *haao-1(tm4627)* animals at 20°C and at two different ages. In all cases, the *haao-1(tm4627)* and 3HAA treated worms resemble worms treated with *haao-1(RNAi)*.

Line 93, related with Figure 1A and Extended Figure 3 and 4: “Like *kynu-1(RNAi)* but unlike *tdo-2(RNAi)*, this benefit was consistent across the viable temperature range for *C. elegans*”. This assumption seems clear in Figure 1A. However, it seems that lifespan extension is much stronger at 15°C than at 25°C in extended figures 3 and 4. Would it be possible to quantify the impact of temperature on *haao-1(RNAi)*-mediated lifespan extension from these extended panels? How the authors reconcile these two different observations?

We took a closer look at all of our *haao-1(RNAi)* data to see what trends were consistent across experiments. The difference in the shape of the lifespan curves at 15°C and 25°C (which is consistent across many experiments and interventions, including and beyond those shown in this manuscript) makes this comparison somewhat imprecise – interventions tend to more consistently extend both early and late lifespan at 20°C and 15°C, while preferentially extend late lifespan at 25°C. This is certainly true for our data on *haao-1(RNAi)*, as well as *kynu-1(RNAi)* and *tdo-2(RNAi)*. With that caveat in mind, the lifespan extension from *haao-1(RNAi)* is consistently slightly larger at 15°C and 20°C (~+30%) than at 25°C (~+25%) across all of our experiments. The difference is a bit larger for the experiments in Extended Data Figures 3 and 4 (+24% vs. +32%) than for the experiments in Figure 1 (+27% vs. +32%), as the reviewer notes. The temperature discrepancy is much larger for *tdo-2* (+6.5% vs. 26.2%). We adjusted the text to make this point explicit. Lifespan experiments are, generally speaking, quite noisy, and this amount of variation between experimental replicates in this work is not atypical.

Line 197-198, related to Figure 3: “Indeed, lifespan extension from *haao-1(RNAi)* was absent in *C. elegans* with the hypomorphic *skn-1(zj15)* mutation (Fig. 3G)”. Very interesting data. Following the experimental design of figure 3, 3HAA supplementation shouldn't extend lifespan in *skn-1(zj15)* animals. Likewise, lifespan extension in the null mutation *haao-1(tm4627)* should be absent in *C. elegans* with the hypomorphic *skn-1(zj15)* mutation. Please, validate these sets of experiments.

This point was also made by Reviewer 1. Please refer to our response to Review 1, Point 8 above.

Line 209-217, related to Figure 4 and Extended Figure 8: The authors show mild physiological benefits on aged mice after 3HAA supplementation. Since Rotarod, Grip Strength, immune populations, blood glucose, TG, etc.. were recorded both at baseline and after treatment, I kindly suggest calculating the effects of 3HAA as the %change

from baseline at individual level. This analysis will determine if 3HAA-mediates a beneficial effect on the several metrics of health. Please show these representations for the panels included in figure 4 and extended figure 8.

Line 211-212: Please normalize the latency to fall per gr of body weight. Do the differences remain?

Line 217, related to extended data 8B: Please analyze glucose-clearance trajectories as %change from time 0.

These several points about the way the mouse health metrics was measured are all good (analyzing percent change, correcting for body weight, and examining the glucose from baseline). We have reworked the data in this section as the reviewer suggests. In some instances, this eliminated the impact of 3HAA (suggesting that the original observed difference was a technical artifact), and in other cases it did not. With respect to the change from baseline vs. the change at each time point, there were cases where some mice did not survive to the second measurement point. The difference between the two analysis approaches may result from survival bias when all mice are included, or it may remove significance due to fewer mice being included when the data is examined as a change from baseline. We note a couple of places where this results in different interpretation in the text.

Line 245, related to statistical significance: the authors claim that dietary 3HAA promotes lifespan extension based on a Cox-regression method to violate the proportional hazard assumption. However, for the specific experimental set/design, in which mice are supplemented with 3HAA late in life (27mo old), a proportional hazard ratio (i.e. same risk of dying) should be in my opinion applied and/or assumed. In lines 466-467, it is not clear to me whether the 'failure' of the Cox proportional hazard regression model means that the statistical test could not be applied or, on the other hand, if the test was run and statistical significance was not found. In addition, the estimation of the survival function/time cannot be achieved with this model. Therefore, I am kindly asking to provide the survival time and statistical evaluation using the Kaplan-Meier model

This is indeed a complicating limitation to this particular mouse study, and likely a result of our small sample size. For context, the study was conducted as a pilot through the Jackson Laboratory Nathan Shock Center and we were limited to the mice that were available at the time. At the reviewers request, and because we suspect that other readers will want to see them, we conducted a post-hoc analysis using the log-rank test and include the results in Table S16 for reference. However, we do not consider these results to be interpretable (see detailed discussion below). Also, because this request asks that multiple statistical tests be run on the same data, we felt obligated to adjust all p values in these experiments for multiple correction considering both analyses. All significance values now reflect this multiple test correction.

As to our methodology, we selected Cox Proportional Hazard regression as a standard method for detecting differences in the hazard function. Because this test makes the

assumption that the compared groups will experience proportional hazard functions, we sought to test this assumption before running the test. We decided in advance to use an alternative statistical test that does not assume proportional hazards in the event that the assumption was violated. For clarity, the proportional hazard assumption does not mean that all individuals have the same risk of dying (i.e., risk of dying), but that the ratio of hazards between the test groups remains constant over time. We do not see that this is obviously true for 3HAA supplemented vs. non-supplemented animals, particularly given that the intervention started very late in life (27-28 months), and thus included a test of this assumption as part of our statistical approach. While Cox PH regression can be robust to violations of the proportional hazards assumption, this is only true for large sample sizes. Because this study was a pilot and included a small samples size in each test group, confirming the assumption of our statistical test is particularly important in this case. Since the assumption failed, we followed our initial strategy and resorted to an alternative, albeit less commonly employed, statistical test that does not make the proportional hazards assumption. We continue to rely on this approach in the current manuscript, though as noted above, the requested post hoc log rank test p values are reported for reference and all values adjusted for multiple tests).

Because we were constrained by the sample size and this experiment was designed as a pilot study, we were careful to not strongly interpret the positive result and explicitly note that these experiments will need to be replicated before they can be considered strong evidence for lifespan extension from 3HAA supplementation in mice. In summary, we feel strongly that presenting the statistical approach originally selected at the outset of the study, and consequently limiting the strength of our interpretation until validation experiments can be run, is the most appropriate and honest way to proceed. Switching to an interpretation based on a post-hoc statistical test for which the assumptions are explicitly violated by the data would leave us unable to arrive at any conclusion (strong or weak).

To the final request, the Kaplan-Meier model was used to estimate the survival function (i.e., generate a survival curve) in all cases and does not itself provide means for statistical comparison between groups. The survival curves provided in Figure 4A were generated using the Kaplan-Meier method.

Finally, in the time that it has taken to collect the data to answer the critiques in this set of reviews, we completed a second pilot lifespan study on *Haa0* knockout mice. We followed the same statistical approach laid out for the 3HAA study, but in this case the survival data did not violate proportional hazards assumption, making the log rank test statistically valid. Like the 3HAA study we did observe that the *Haa0* knockout mice lived longer than wild type mice. This difference was significant in females, with a trend that did not reach significance in males. We now include this data and associated health metrics in Figure 4 and Extended Data Figure 9. This study was also relatively small (15-25 mice per genotype per sex), but we hope that the addition of this study adds merit to the mouse work presented.

Line 245: the authors claim that dietary 3HAA is sufficient to promote healthy longevity in the mouse model. How the authors explain this effect? How is 3HAA metabolized? Are the levels of 3HAA increased? Are the levels of other tryptophan-metabolites modulated? Does 3HAA supplementation reduce oxidative stress and activates cellular Nrf2 response? All these questions remain unresolved. In my opinion, the activation of the main 3HAA-mediated mechanisms should be proved, especially when the sample size for these set of experiments is very small. In the end, the authors should demonstrate how 3HAA is acting in rodents.

We agree that these are important questions and that they are currently unresolved. We now include blood and urine metabolite data demonstrating that physiological 3HAA levels are substantially upregulated in response to both dietary 3HAA and *Haao* knockout. We also present data for the other metabolites that were affected in *C. elegans*. We are very interested the mechanisms of 3HAA mammalian aging, but a full investigation is beyond the scope of the current manuscript. As we note above, the 3HAA diet study was a small longitudinal study conducted as part of a Nathan Shock Center pilot by The Jackson Laboratory. The newly added *Haao*^{-/-} experiment was similarly a small longitudinal study carried out as a pilot study with startup funds. Because they were intended as a first look at lifespan, we only collected a small amount of plasma and urine, as well as the reported behavioral work. We were unable to collect additional tissue because the endpoint was lifespan. The plasma and urine samples were used to measure kynurenine metabolites and the few other traits reported. We do not currently have access to additional tissue, plasma, or urine samples to conduct further analyses.

This work is currently unfunded, and we do not have the resources to conduct a detailed mechanistic study in mice at present. We have applied for funding to carry out these studies and have been unsuccessful in part because the current work remains unpublished (thus creating a chicken-or-the-egg problem). Our purpose in this work was to present the *C. elegans* studies as a primary focus and include the mouse work to demonstrate that the primary physiological outcomes (particularly increased lifespan) appear to be translatable to a mammalian system. We again emphasize that this comes with the caveat that the studies were conducted on a small number of mice and should be taken as tentative until fully powered studies can be conducted. We ask the reviewers to consider the current mouse work as-is given the circumstances.

Additional recommendations:

Line 94-95, relative to Figure 1C: the authors indicate that *haao-1*(RNAi) animals were slow to reach full body size (Fig 1G), but resistant to the age-dependent reduction in body size observed in animals subjected to empty vector (EV) RNAi. I disagree with this comment since *haao-1*(RNAi) never reaches full body size (it is lower than controls at 7, 14 and 21 days at 15°C; and also lower than controls at every time-point analyzed at 25°C]. It would be helpful to understand if *haao-1*(RNAi) and/or 3HAA supplementation impacts well-characterized metabolic pathways that governs cell growth (mTOR

pathway, AMPK,...).

The *haao-1* deficient animals do reach a maximum size at 15°C around day 21 that is similar to the maximum size for wild type animals at day 14. At 25°C they are not significantly smaller than wild type at days 4 and 8, but we agree that the maximum size for wild type animals (day 11) is greater than the maximum observed size for *haao-1* deficient animals. We now make this a bit clearer in the Results text. The *Haao* knockout mice also have modestly reduced body weight early in life, which raises a similar question. We have not exhaustively looked at interactions with growth pathways, but *haao-1(RNAi)* did not further extend lifespan of the *rsks-1* mutant animals, suggesting a possible interaction with the translational arm of mTOR signaling. Lifespan extension from *haao-1(RNAi)* is also partially blocked by deletion of *daf-16* at both temperatures. Since the insulin pathway mediates growth associated with dauer formation and dauer-like states, this is another possible candidate. We will be examining cellular growth phenotypes as part of an ongoing study of 3HAA in cancer and plan to take a broader look at growth pathways once we are able to obtain funding for a more in depth aging study that includes tissue collection as part of a larger cross-sectional cohort. We note the impact on growth as a potential limitation to therapeutic elevation of 3HAA in the last paragraph of the Discussion.

Line 99-100, related to Fig. 1E: “Total brood size was not impacted by *haao-1(RNAi)* (Fig. 1E); however, egg production was slightly delayed relative to EV(RNAi), similar to *kynu-1(RNAi)* and less pronounced than *tdo-2(RNAi)* (Extended Data Fig. 1A)”. In my opinion, caution should be used here if these enzymes/metabolites impact fertility. Please extend the discussion of this topic accordingly, which confronts with the potential role of HAOO and 3HAA as novel therapeutics targets to combat aging. Including these negative-side effects on the abstract is recommended.

Good point. We have added this to our discussion of potential limitation of 3HAA as a therapeutic target (last paragraph of Discussion).

Line 206-209, related to Fig 4A: In my opinion, Fig 4A-top panel is, in my opinion, not accurate, since it seems that mice were randomized into the experimental groups at birth and any of them died birth to at least 27 months of age. Rather, mice at the age of 27 months (survivors) were divided into the experimental groups. I would recommend to remove Fig 4A-bottom panel and incorporate age (in weeks) together with Time on diet (in weeks) on Fig4A-bottom panel.

We showed the data both from birth and from the start of the diet to give a sense of the impact relative to full lifespan, in part provide some conservative context (since the latter chart in some ways visually inflates the effect of the intervention). However, we see the point that the data is not accurately portrayed, since we are not showing all potential mice that started from birth and did not make it to the time when the intervention was initiated. We removed the first panel and now show only the survival curves from the start of the dietary intervention.

Line 137-140: "If our model is accurate and 3HAA mediates benefits of decreasing HAAO activity, but not KYNU or TDO2, then 3HAA should further increase the already long lifespans of animals with reduced *kynu-1* or *tdo-2*, but not animals with reduced *haao-1*. Our observations confirmed this prediction (Fig. 3F, Extended Data Fig. 7)": I think this data set do not correspond to Figure 3F and Extended Fig 6 do not show that data.

Thank you for catching the error; this was a mis-reference in the original manuscript. In the current version these data are Figure 2H and Extended Data Figure 7. The text should now be correct.

Line 206: please include the number of mice in this sentence (i.e. n = 6-8 mice per group).

Mouse numbers are now indicated in the text and figure legends for the original 3HAA diet study and the newly added *Haao* knockout study.

Lines 257-259: Since the authors highlight that *haao-1*(RNAi), *kynu-1*(RNAi) and *tdo-2*(RNAi) reduces NAD levels, while elevating NAD is shown to be beneficial for healthy aging, could the authors show if acetylation (as a function of reduced activity of the NAD-consuming enzyme Sir) is increased?

We have a separate ongoing project looking at this interaction. We have not looked specifically at the activity of *sir-2.1* or downstream acetylation, but this is a good idea. The connection between kynurenine metabolism and NAD⁺ appears to be more complex than it appears at first glance, and there is probably some cross regulation. Here is a brief discussion of what we are looking at, which we hope to publish in a more complete form in a future manuscript.

The interaction between our kynurenine pathway interventions and NAD⁺ metabolism is one of our most frequently asked questions. We have not focused on NAD⁺ here because, as the review points out, knocking down kynurenine pathway (aka *de novo* NAD⁺ synthesis) enzymes would be expected to shorten lifespan, based on the literature reporting that interventions that increase NAD⁺ extend lifespan. This suggested out of the gate that the mechanism of lifespan extension resulting from kynurenine pathway knockdown was likely to be distinct from the impact on NAD⁺. Because NAD⁺ is modestly repressed in response to knockdown of *tdo-2*, *kynu-1*, and *haao-1*, this suggests first that the kynurenine pathway is not a major source of NAD⁺ in worms, at least in the context of normal nutrition. It also suggests that you may expect synergy between knockdown of a kynurenine pathway enzyme and an intervention that increases NAD⁺ availability. Compiling our examination of this interaction, we have not been able to replicate the published lifespan extension from NAD⁺ precursors in worms. We are troubleshooting this issue. We do find that knocking down a subset of NAD⁺ consuming enzymes enhances the lifespan extension from *haao-1* deficiency. This is not true for *sir-2.1* (in fact, *sir-2.1* knockdown partially represses lifespan extension from

haao-1, as shown in Figure 1G and Extended Data Figures 3 and 4). This suggests a context specific interaction between *haao-1*, 3HAA, and NAD⁺ that we do not yet understand in detail, but continue to investigate.

Reviewer #3 (Remarks to the Author):

The paper by Dang et al. analyzed the effects of knocking down the KP enzyme *haao-1*, and supplementing with 3HAA, the substrate of *haao-1*, in *C. elegans*. Their study uses a mix of phenotypic assays (lifespan, motility, fecundity) and biochemical assays (metabolite MS, H₂O₂, GFP reporters) to conclude that knockdown of *haao-1* increases lifespan via the accumulation of its substrate 3HAA. They show that treatment with 3HAA has an anti-oxidant effect, and activates the oxidative stress response in *C. elegans*. They finally validate their findings in a small cohort of mice to show that 3HAA has a similar effect on lifespan, as well as improvement in health and behavioral functions. The study is of interest to the anti-ageing field as it suggests that 3HAA may be a promising therapeutic target to target the effects of ageing. The manuscript is well-written and the data are clear and convincing. With a few small changes we can recommend this manuscript for publication.

In Figure 2 and Extended Figures 5,6 the authors show evidence that 3HAA supplementation mediates *haao-1* lifespan extension. In extended Data Figure 1B, the *haao-1* RNAi also results in an increase in various other metabolites including tryptophan, kynurenic acid and cinnabarinic acid. A useful additional experiment would be to also supplement with these additional metabolites and perform lifespan assays to further elucidate potential mechanisms of *haao-1* knockdown. Additionally, to further show that 3HAA supplementation can replicate the effect of *haao-1*, it would be fruitful to also check motility assays in response to 3HAA in a separate phenotypic assay (as in Figure 1).

These are both good additions. As we noted in an earlier response, we include an additional kynurenine pathway metabolite experiment at 20°C (see Extended Data Figure 6G-K) and found that the increase in KYN and the reduction in AA in response to *haao-1*(RNAi) replicated at this temperature. We now include lifespan experiments for animals supplemented with both metabolites. KYN slightly shortened lifespan, which is inconsistent with it playing a primary role in lifespan extension, since it was elevated in response to *haao-1*(RNAi). However, AA shorten lifespan in a dose-dependent manner. Since AA was reduced in response to knockdown of *tdo-2*, *kynu-1*, and *haao-1*, reduced AA provides a second potential mechanism for lifespan extension that is common to all three genes (see Extended Data Figure 6D). These results are now included in our Discussion, second paragraph. To the reviewer's latter request, we also now include thrashing and brood size data for worms treated with 3HAA, which follow a similar pattern to *haao-1*(RNAi) (see Figure 2F,G and Extended Data Figure 5B).

In Figure 3 and Extended Figure 7, the authors show that 3HAA boosts oxidative stress resistance. Figure 3B would be improved with some element of quantification, such as a

spectrophotometric assay. It would also be useful to check motility of *haao-1* RNAi in a *skn-1* mutant background (Figure 3G) to see if the effect is the same in an additional phenotypic assay.

This is a good idea and something that we have tried to do. While the color change is obvious visually, quantification has proven challenging. The peak produced by 3HAA on the spectrophotometer is very broad and we have not been able to develop a quantitative assay that correlates well with 3HAA concentration. As noted above, we find that endogenous hydrogen peroxide is actually increased in the worms, which better corresponds to activation of NRF2/SKN-1. This data is now included in Figure 3E.

General notes: many replicate number and sample sizes not clearly indicated for many of the experiments in the main text, figure legends, or materials and methods.

Given the number of *C. elegans* experiments conducted throughout the manuscript and the length limits on the text, we did not list sample numbers for every worm experiment in the main text. These are typically ~200-300 animals for lifespan experiments, and dozens for fluorescent experiments. While the specific sample size is not listed in the main text, we do make a note where appropriate for the general experiment design including animal numbers in the methods and include a full accounting of all sample sizes for all experiments in Supplemental Tables. We do now include sample sizes for the mouse studies in both the text and figure legends, as these are important for interpretation given that the studies were relatively small.

There are a few typos: in Line 144, should be 2F instead of 3F,

Thank you, corrected and updated for the altered figures in the resubmission.

in Figure 4D “should be Inflammatory” instead of “inflammmitory”.

Thank you, corrected.

Some figures duplicated in the main figures and in the extended figures, e.g. Figure 2D and Extended Figure 5C.

This was intentional for the dose response experiments, where it is helpful to have the dose response curve and the corresponding survival curves next to one another for comparison. We included the dose response in the main figures due to space constraints.

REVIEWERS' COMMENTS

Reviewer #1 (Remarks to the Author):

Authors have addressed my queries, including new experimental data regarding the proposed mechanism. Certain points have been discussed, and claims have been fine-tuned to align more closely with the supporting data.

I have no further comments.

Reviewer #2 (Remarks to the Author):

The revised version of the manuscript has now incorporated nearly all of additional experiments requested and recommendations that were included, significantly improving the quality of the manuscript. Few additional recommendations are raised on this new version:

1- In Figure 4, the authors now show that female haa0-KO mice, but not male, live longer than WT. In this context, and taking into consideration the relevance of sex-specific differences in the field of aging, I recommend to discuss this topic in the discussion section, and/or to discuss why 3HAA supplementation did extend lifespan also in male mice.

2- In Figure 4, panel A, "Colored text indicates change in mean remaining lifespan relative to control". The magnitude of this change seems different between the previous [+75.1%, blue; +48.4%, red] and current [+65.8%, blue; +39.7%, red] versions of the manuscript. Please revise the analysis.

3- The authors have clearly indicated in the rebuttal letter that funding constitutes a primary limiting factor to address how dietary 3HAA is sufficient to promote healthy longevity in the mouse model. Likewise, they clearly stated that i) mice "experiment was designed as a pilot study" and "included a small samples size in each test group", ii) "we were careful to not strongly interpret the positive result" and iii) "these experiments will need to be replicated before they can be considered strong evidence for lifespan extension". These statements are aligned with my initial peer-review, and I definitively agree that lifespan and healthspan studies in the mouse model cannot be considered strong evidence. Therefore, I strongly encourage to include the terms "a pilot study" both in the title and abstract when the

mouse model is referenced.

Reviewer #3 (Remarks to the Author):

We are satisfied with the revisions by the authors.

RESPONSE TO REVIEWER COMMENTS

We thank the referees for taking the time to give our manuscript and additional round of review and for their support of this work. We have addressed each of the outstanding critiques and comments in our new revision. We address each specific point directly below.

Reviewer #1 (Remarks to the Author):

Authors have addressed my queries, including new experimental data regarding the proposed mechanism. Certain points have been discussed, and claims have been fine-tuned to align more closely with the supporting data. I have no further comments.

Thank you for your review and support.

Reviewer #2 (Remarks to the Author):

The revised version of the manuscript has now incorporated nearly all of additional experiments requested and recommendations that were included, significantly improving the quality of the manuscript.

Thank you for your comments and attention to detail; the manuscript is much improved from your efforts. Please see responses to your last few comments below.

Few additional recommendations are raised on this new version:

1- In Figure 4, the authors now show that female haao-KO mice, but not male, live longer than WT. In this context, and taking into consideration the relevance of sex-specific differences in the field of aging, I recommend to discuss this topic in the discussion section, and/or to discuss why 3HAA supplementation did extend lifespan also in male mice.

This is a good suggestion and we added a few sentences to the discussion that outline potential differences between the 3HAA diet study and the Haao KO study. In particular, there are a number of variables that we don't yet understand that may contribute to the differences between the study, such as Haao KO resulting in a different tissue distribution and concentration of 3HAA (plasma values were much higher for the KOs than the diet study), the likelihood that we have not yet optimized dosing or delivery, and variation that may arise from the environment (the diet study was conducted at The Jackson Laboratory, while the KO study was conducted at the University of Arizona). We also note that therapeutic strategies using Haao inhibitors may be used to more precisely tune these parameters in the future.

2- In Figure 4, panel A, “Colored text indicates change in mean remaining lifespan relative to control”. The magnitude of this change seems different between the previous [+75.1%, blue;+48.4%, red] and current [+65.8%, blue; +39.7%, red] versions of the manuscript. Please revise the analysis.

At the time of the original submission there was still one mouse on the control diet alive. The revision included this mouse and that shifted the % change in survival. Apologies for our error in not indicating the change in a comment in the revision document. The update does not change the conclusions or analysis. Incidentally, the same situation occurred with the *Haa0* knockout study (the last mouse died between revisions), and we have accordingly updated our statistics.

3- The authors have clearly indicated in the rebuttal letter that funding constitutes a primary limiting factor to address how dietary 3HAA is sufficient to promote healthy longevity in the mouse model. Likewise, they clearly stated that i) mice “experiment was designed as a pilot study” and “included a small samples size in each test group”, ii) “we were careful to not strongly interpret the positive result” and iii) “these experiments will need to be replicated before they can be considered strong evidence for lifespan extension”. These statements are aligned with my initial peer-review, and I definitively agree that lifespan and healthspan studies in the mouse model cannot be considered strong evidence. Therefore, I strongly encourage to include the terms “a pilot study” both in the title and abstract when the mouse model is referenced.

We also support the effort to be as precise as possible in describing the study and making claims about the data. We changed the title from the previous declarative statement to “On the benefits of the tryptophan metabolite 3-hydroxyanthranilic acid in *Caenorhabditis elegans* and mouse aging”. This is somewhat of a throwback to how scientific articles used to be titled, and one that we have been considering adopting for papers more broadly. This approach better reflects the subtleties of the data and does not make definitive claim about the effects of 3HAA that might be misrepresented in references by simply quoting the title. We also noted that the mouse studies were both pilot studies in the revised abstract and the text.

Reviewer #3 (Remarks to the Author):

We are satisfied with the revisions by the authors.

Thank you for your review and support.